# Serum Proteomics Reveals Diagnostic Biomarkers and Molecular Pathways in Cerebral Palsy

Yiran Xu[1,13], Chi Ma[2,13], Yanyan Sun[3,13], Jiajun Zhu[2,13], Shiman He[2,13], Hui Gao[1], Subei Tan [2], Lingling Zhang[1], Jinwen Feng [2], Yangong Wang[4], Sha Tian [2], Qinghe Xing[4], Jiamei Zhang[1,5], Yanan Wu[1], Xiaoli Zhang[1], Lirong Zhang [6], Dengna Zhu[1,5], Michael Kruer[7,8], Xiaoyang Wang [1,9], Jozef Gecz [10], Changlian Zhu [1,11,12] ✉ & Chen Ding [2] ✉

Cerebral palsy (CP), a prevalent non-progressive neurological disorder in children, lacks reliable biomarkers for early diagnosis, and its molecular mechanisms remain poorly understood. In this study, we conducted serum proteomic profiling of 346 CP patients and 190 healthy controls and developed a 10-protein multi-marker panel for application in diagnosis of CP. The panel was further validated in an independent CP cohort using an orthogonal method, enzyme-linked immunosorbent assay (ELISA). By integrating serum proteomic data with whole-exome sequencing (WES) results, we found that CP patients carrying pathogenic variants exhibited downregulation of synaptic and calcium signaling pathways at the protein level. We also explored the impact of clinical risk factors on the proteome, identifying disruptions in lipid metabolism associated with low birth weight and low gestational age. Additionally, we found a positive correlation between Immunoglobulin Heavy Variable (IGHV) families and higher Gross Motor Function Classification System (GMFCS) levels. Overall, our study provides a valuable tool for early CP diagnosis that complements standard clinical and genomic assessments, and suggests potential molecular mechanisms associated with CP pathogenesis, highlighting the interplay among genetic, environmental, and protein network factors.

Cerebral palsy (CP) is a nonprogressive neurodevelopmental disorder (NDD) affecting motor function and brain development, and is often associated with communication disorders, sensory deficits, cognitive deficits and epilepsy[1]. It has a global prevalence of 1.6‰, with higher rates in developing countries[2–4]. Based on the predominant type of movement disorder, CP is categorized into spastic, dyskinetic and ataxic CP[5]. Among these, spastic CP is the most common, representing approximately 80% of cases[6]. The Gross Motor Function Classification System (GMFCS: levels I–V) has become the gold standard to evaluate the severity of motor disorders with CP[7]. The pathogenesis of CP is

complex and involves both genetic and environmental risk factors, which pose a great challenge for clinical diagnosis and the study of underlying pathological mechanisms.

Early diagnosis of CP maximizes neuroplasticity and minimizes harmful effects on muscle and bone development[5]. In clinical practice, CP diagnosis typically involves a combination of neurological assessments, developmental evaluations, and neuroimaging techniques such as magnetic resonance imaging (MRI)[5,8]. Although MRI enhances diagnostic sensitivity by revealing structural abnormalities, approximately 10% of CP patients show normal MRI findings, underscoring the

limitations of relying on imaging alone[5]. Given these challenges, there is increasing interest in molecular biomarkers. Serum biomarkers are capable of detecting disease-specific alterations and have shown diagnostic utility across various neurological disorders[9,10]. However, their application in CP remains limited, highlighting the need for further studies to identify and validate serum biomarkers for early diagnosis.

Both clinical risk factors and genetic variants are well-established contributors to CP etiology[11–14]. Preterm birth (gestational age of less than 37 weeks) and low birth weight (less than 2500 g) are recognized as major risk factors for CP[15]. Notably, preterm birth and low birth weight are particularly associated with spastic CP, possibly due to injury to the corticospinal tracts and periventricular white matter regions critically involved in spastic CP pathophysiology[16]. However, the underlying molecular mechanisms linking these factors to CP remain largely unexplored, limiting the development of patient-stratified therapies. In addition to clinical risk factors, genetic factor also plays an important role, with approximately 30% of cases linked to genetic disorders[14]. In a previous study, we performed a large-scale analysis of genetic variants in Chinese patients with CP, identifying an enrichment of CP-associated variants in pathways related to synaptic function and ion transport[17]. Despite these findings, the impact of these pathogenic variants on downstream protein expression remains largely unexamined. To address these gaps, a comprehensive analysis integrating genomic and proteomic data is needed to elucidate the molecular consequences of genetic and clinical risk factors in CP.

To identify early diagnostic biomarkers and elucidate protein alterations associated with genetic and clinical risk factors, we conducted an integrative multi-omics study combining serum proteomics (CP = 346, healthy control = 190) and whole-exome sequencing (WES) data from 321 individuals[17]. We constructed a multi-marker predictive model based on proteomic data and validated its diagnostic performance using enzyme-linked immunosorbent assay (ELISA) in an independent cohort (CP = 38, healthy controls = 32). In addition to developing the predictive model, we explored the effects of genetic variants and clinical risk factors on the serum proteome. Notably, weighted gene co-expression network analysis (WGCNA) revealed a strong association between disease severity and inflammatory protein modules. Collectively, our study offers a valuable resource for advancing our understanding of CP's molecular landscape and improving diagnostic strategies.

## Results
### Serum proteomic profiling from cerebral palsy
To provide a systematic and comprehensive proteomic characterization of CP, we collected the serum samples from 346 CP patients, including dyskinetic subtype ($n = 17$), spastic subtype ($n = 278$), ataxic subtype ($n = 9$), mixed subtype ($n = 40$), and 190 healthy controls (HC) (Fig. 1A). The demographic and clinical information of CP patients, including clinical subtypes, age, gender, GMFCS, MRI result, etc., were shown in Fig. 1B and Supplementary Fig. 1A, B (Supplementary Data 1). GMFCS has been used to evaluate disease severity, including level I (26.2%), level II (24.4%), level III (17.4%), level IV (13.7%), and level V (18.3%). The MRI Classification System (MRICS) was used to categorize brain MRI findings into five types[18]: Maldevelopments (6.1%), Predominant white matter injury (61.8%), Predominant gray matter injury (14.5%), Miscellaneous (9.8%), and Normal (7.8%). The median ages of the CP and HC groups were 23 and 24 months, respectively. Notably, no significant differences were observed between CP and HC in age ($p = 0.16$) or sex distribution ($p = 0.233$) (Supplementary Fig. 1A).

We employed a mass spectrometry-based, high-throughput data-independent acquisition (DIA) quantitative proteomics approach to identify serum proteins[19–21] ("Methods"). To ensure quantitative reproducibility and assess instrumental stability throughout the LC–MS/MS workflow, pooled serum samples and HEK293T cell samples were used as quality control (QC) samples[21–24] ("Methods"). The average Spearman's correlation coefficients were 0.97 for pooled QC samples and 0.98 for 293 T QC (Supplementary Fig. 1C). Further analysis revealed that the coefficients of variation (CVs) for the pooled QC and 293 T QC samples were 17% and 18%, respectively (Supplementary Fig. 1D). These results suggested the consistent stability of the entire workflow and MS platform.

An average of 1994 and 2019 identified proteins per CP and HC serum samples, respectively (Supplementary Fig. 1E). Our proteomics data exhibited a unimodal distribution and the quantified protein intensities spanned over eight orders of magnitude (Supplementary Figs. 1F and Supplementary Fig. 1G). Ultimately, 2050 serum proteins were selected for downstream analysis after data filtering (Supplementary Data 1).

### Proteomic alterations in CP compared to healthy controls, revealing candidate biomarkers
To gain insights into serum protein alterations in patients, we compared the serum proteome profiles between CP children and HC (Supplementary Data 1). We found 82 proteins were significantly increased (e.g., ARHGEF10, ADAMTSL4) (FDR < 0.05, CP/HC, Fold Change > 1.5) and 83 proteins were significantly decreased in CP (e.g., UBE2A, ADH1B) (FDR < 0.05, CP/HC, Fold Change < 0.67) (Fig. 1C). The upregulated proteins in CP were significantly enriched in ECM (e.g., AGRN, VWF, and FN1), cell adhesion (e.g., SELL, BCAR1, and ITGAM) and sugar metabolism pathways (e.g., HK2 and HEXA) (FDR < 0.05). Conversely, proteins decreased in CP were enriched in calcium signaling (e.g., STIM1, ITPR3, GNAS) and thyroid-stimulating hormone pathways (e.g., APEX1 and STAT3). These findings suggested systemic dysregulation in ECM organization, cellular signaling, and metabolic pathways.

To leverage this data for the development of diagnostic models, we randomly divided 536 serum samples into a training set ($n = 429$) and a testing set ($n = 107$) at a ratio of 4:1. To prevent information leakage and ensure model generalizability, diagnostic protein selection was performed exclusively within the training set using the Extreme Gradient Boosting (XGBoost)[25] (Fig. 1D; "Methods"). The top 10 most informative proteins were selected for model construction, comprising six upregulated proteins (DHX9, CUTA, LONP1, BCAR1, SPARC, and ARHGEF10) and four downregulated ones (ANXA2, MME, GNAI3, and MANBA) in CP (Fig. 1E). Notably, their differential expression between CP and controls remained statistically significant after adjusting for age and sex (Supplementary Fig. 1H; "Methods"). Based on these 10 biomarkers, we constructed a multi-marker diagnostic model that achieved high performance in the testing set (AUC = 0.96) (Fig. 1F). We further assessed model robustness to outliers ("Methods"). After correcting for outliers, the 10-biomarker predictive model exhibited stable performance (AUC = 0.96), further supporting the reliability of the selected biomarkers and their potential for clinical application (Supplementary Fig. 1I).

Given that spastic CP accounts for approximately 80% of all cases, we evaluated whether subtype composition might influence diagnostic performance. To this end, 66 spastic and 66 non-spastic CP patients were randomly selected, along with 190 HC samples. Using the XGBoost algorithm with 10-fold cross-validation, the 10-protein diagnostic model showed robust diagnostic performance, achieving an AUC of 0.91 (Supplementary Fig. 1J). These findings suggested that the diagnostic performance of the multi-marker model remained consistent across CP subtypes, supporting its potential applicability in clinically heterogeneous CP populations.

Previous studies have reported that some CP patients exhibit normal MRI findings, which can complicate clinical diagnosis[5,15]. To assess whether our 10-protein diagnostic model could reliably identify CP patients with normal MRI results, we further analyzed this subgroup within our cohort. In our cohort, 273 patients presented with abnormal

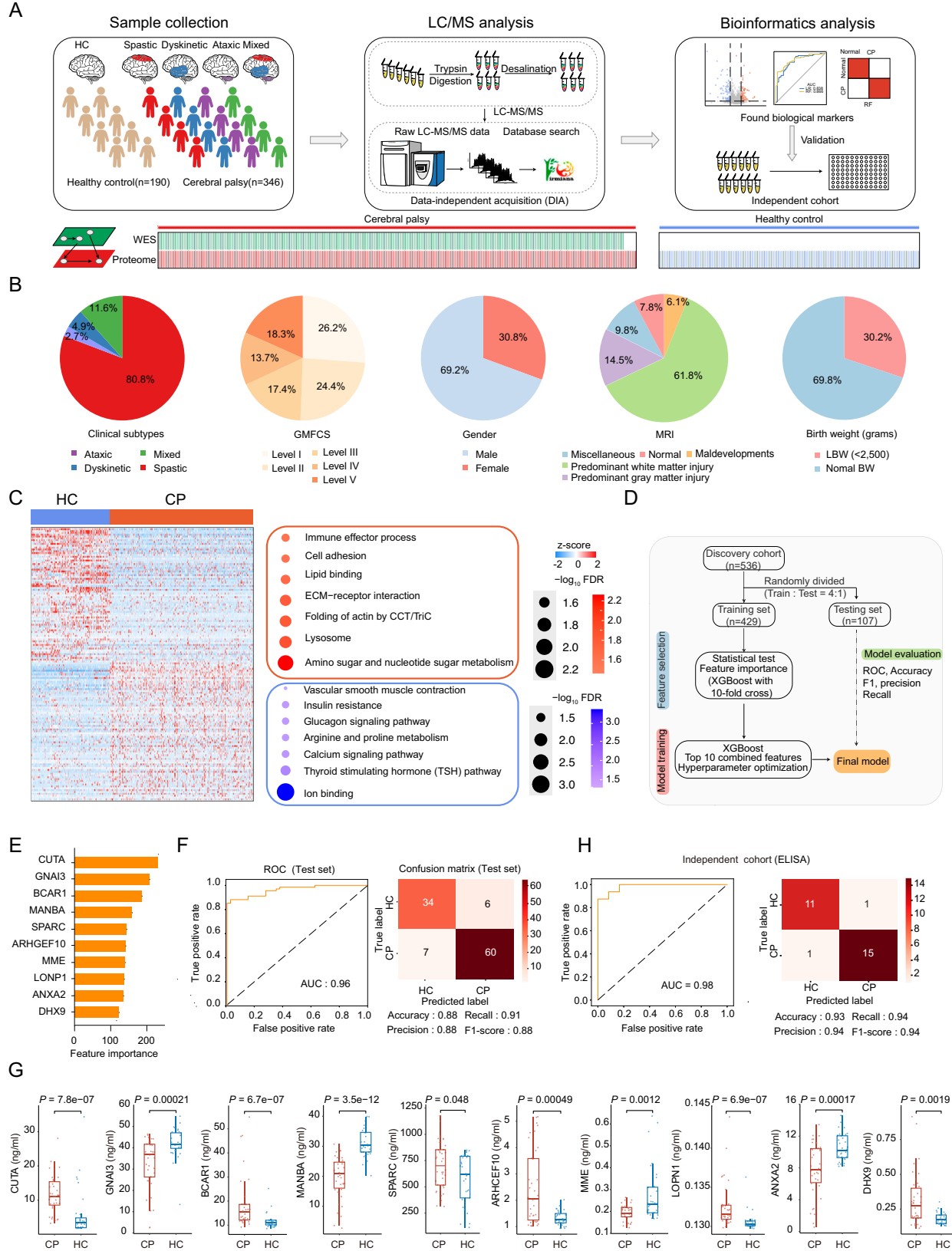

MRI findings, while 23 had normal imaging results (Supplementary Fig. 1B). A balanced cohort was constructed by pairing the 23 MRI-normal CP patients with 23 randomly selected HC samples. In this subgroup, the 10-protein diagnostic model achieved an AUC of 0.86 (Supplementary Fig. 1K), suggesting that the diagnostic panel retained performance in the absence of structural MRI abnormalities. Further

validation in larger MRI-normal cohorts is needed to confirm these findings.

To evaluate the cross-platform predictive performance of the 10 candidate proteins, we measured their expression levels using ELISA in an independent cohort comprising 38 CP patients and 32 HC samples. All 10 proteins showed significantly altered expression in CP,

**Fig. 1 | Study design and alterations in the serum proteome between children with cerebral palsy and healthy controls. A** The proteomics workflow outlining three modules: sample collection, LC/MS analysis, and data analysis. **B** Pie chart showing clinical information of CP samples, including clinical subtypes, MRI, gender, GMFCS, and birth weight. **C** Heatmap and Bubble plot illustrating distinct proteins and pathways differentiating CP ($n = 346$) from HC ($n = 190$). The bubble size corresponds to the significance of the pathways enriched in each group. The red bubbles indicate CP, and the blue bubbles indicate HC. The two-sided Wilcoxon rank-sum test was used for differential analysis, with *p*-values adjusted for FDR. **D** The schematic showing the machine learning framework used to develop classifiers for identifying CP and healthy control samples. **E** The feature importance in distinguishing CP from healthy controls using the XGBoost-based model. **F** The Receiver operating characteristic (ROC) curve showing the performance of the multi-biomarker model in the testing set based on proteomic results, and the confusion matrix showing the performance of the ML classifier. **G** Boxplots showing 10 diagnostic protein differences between CP ($n = 38$) and healthy control ($n = 32$) in the validation cohort using ELISA (two-sided Wilcoxon rank-sum test). Boxplots show median (central line), upper and lower quartiles (box limits), 1.5× interquartile range (whiskers). **H** The ROC curve showing the performance of the multi-biomarker model in the testing set based on ELISA results, and the confusion matrix showing the performance of the ML classifier. Source data are provided as a Source Data file.

consistent with the proteomic results (Fig. 1G). Based on the ELISA data, we constructed a 10-protein diagnostic model using XGBoost with 10-fold cross-validation, which achieved excellent performance (AUC = 0.98) (Fig. 1H; "Methods").

To facilitate clinical applicability, a risk score was generated using logistic regression based on ELISA data, which significantly differentiated CP from HC ($p < 0.05$) ("Methods", Supplementary Fig. 1L). An optimal cutoff value of 0.59 was determined using the Youden index (0.95). To improve cost-efficiency and simplify clinical implementation, we further explored whether a smaller subset of proteins could retain comparable diagnostic performance. We systematically evaluated all 120 possible three-protein combinations and calculated risk scores for each. Among all 120 three-protein combinations, the panel of SPARC, DHX9, and MANBA achieved the highest Youden index (0.88) with an optimal cutoff value of 0.49 (Supplementary Fig. 1L). In summary, these findings indicated the cross-platform reproducibility of the 10-protein model and suggest that a simplified three-protein panel might serve as a practical alternative with comparable performance.

## Proteogenomic analysis identifies dysregulation of calcium and synaptic pathways in genetic cerebral palsy at the protein level

Given the distinct proteomic signatures identified in CP, we next examined whether these alterations were linked to specific genetic variants that might underlie the disease. To elucidate the genetic contribution, we integrated serum proteomic data with matched WES data from 321 CP patients ("Methods", Supplementary Data 1)[17]. Given their established clinical relevance, pathogenic and likely pathogenic (P/LP) variants are frequently implicated in disease susceptibility[26]. We therefore focused on these variants and identified 76P/LP variants across 65 patients, some of whom carried more than one variant (Fig. 2A). Notably, variants in *ATP2B3* (c.T2672C: p.M891T; c.A2209G: p.N737D) and *WDR62* (c.C3406T: p.R1136X; c.C1684G: p.H562D) were identified in the spastic subtype. Variants in *GALC* (c.G2041A: p.V681M; c.G1912A: p.g638S) and *ARSA* (c.G938A: p.R313Q; c.T746C: p.F249S) were observed in the mixed subtype. Ataxic cases carried variants in *DNMT3A* (c.1249delT: p.S417Lfs234), *CTNNB1* (c.1789dupC: p.L598Ifs11), and *CEP290* (c.6012-2 A > G; c.4819dupA: p.M1607Nfs*19). In the dyskinetic subtype, we observed variants in *PCDH12* (c.G1067A: p.W356X) and *ITPR1* (c.C805T: p.R269W).

To elucidate the biological relevance of these genes with LP/P variants in our cohort, we performed pathway enrichment analysis, which revealed significant associations with ion binding, synaptic-related signaling, WNT signaling, and chromatin organization pathways (Fig. 2B). Variants identified in the calcium signaling pathway included *CACNA1A*, *CAMK2G*, *CACNA1D*, and *CACNA1G*, which play a crucial role in regulating neurotransmitter release and muscle contraction[27]. Similarly, variants such as *EHMT1*, *DNMT3A*, *CHD4*, *EZH2*, *NSD1*, and *SMARCA4* were enriched in the chromatin organization pathway, which regulates DNA accessibility and transcriptional activity[28]. Furthermore, *CTNNB1* and *TCF4* variants were associated

with WNT signaling, a pathway critical for embryonic development and tissue homeostasis[29].

To assess whether the genetic etiology of CP in our cohort, i.e., individuals with P/LP variants in known disease genes, impacts their proteome, we stratified the cohort into genetic CP, non-genetic CP, and HC subgroups ("Methods"). Comparative analysis among these groups identified 663 dysregulated proteins (FDR < 0.05) (Fig. 2C). Among these, 177 proteins displayed the highest expression in the genetic CP group and the lowest in HC, while 167 proteins exhibited the opposite trend (Fig. 2D). Pathway enrichment analysis revealed that the upregulated proteins were significantly enriched in protein binding (e.g., EFTUD2, CSNK2A3, and EIF3L) and RAC1 GTPase cycle pathways (e.g., RAC1, DOCK10, and CDC42BPA) (Fig. 2E). Downregulated proteins exhibited significant enrichment in signaling pathways, including dopaminergic synapse (e.g., CALM3, ITPR3, and GNAI3), and calcium signaling (e.g., STIM1, CAMK1D, and MYLK) (Fig. 2E). Notably, there was potential concordance between the downregulated pathways and variant-enriched pathways. For instance, in the calcium signaling pathway, the downregulated proteins ITPR3, CAMK1D, and CALM3 were linked to calcium-related variants such as *CACNA1A* and *CACNA1G*. Similarly, in synaptic signaling, the down-regulation of GNAI3 protein correlated with synaptic-related variants, such as *GNAO1* and *GABBR2* (Fig. 2F).

Moreover, according to single-sample gene set enrichment (ssGSEA) ("Methods"), there was a negative correlation between dopaminergic synapses and skeletal muscle signaling pathways with GMFCS (Fig. 2G). This suggested that downregulation of these pathways may impair neurological and muscular functions, potentially contributing to increased motor dysfunction. In summary, our integrated genomic and proteomic analysis revealed that the genetic variants were predominantly enriched in calcium and neural signaling pathways, which were associated with alterations in calcium- and skeletal muscle-associated serum protein expression. Further validation in larger and genetically diverse cohorts is warranted.

## Proteogenomic analysis reveals molecular alterations associated with clinical risk factors

In addition to genetic contributions, we evaluated the impact on the proteome in view of known clinical risk factors of CP. Established clinical risk factors affecting neurodevelopment include preterm delivery, perinatal asphyxia, and intrauterine infection[30,31]. However, there is limited knowledge of the association of these risk factors with genetic variants and proteins. We first categorized the CP samples into several risk subtypes based on previous reports[11]: preconception (e.g., maternal adverse pregnancy history), prenatal (e.g., threatened abortion and pregnancy complications), perinatal (e.g., low birth weight and preterm birth), postnatal (e.g., pathological jaundice), and unknown (no risk factors) (Fig. 3A).

LP/P variants were identified across all clinical risk subtypes (Fig. 3B). In the preconception group, representative variants included *PCDH12*, *EHMT1* and *CACNA1A*. The prenatal subtype was characterized by variants such as *ARSA*, *GALC*, *CTNNB1*, *TCF4*, *ALS2*, and *GABBR2*, which were enriched in sphingolipid metabolism, WNT signaling,

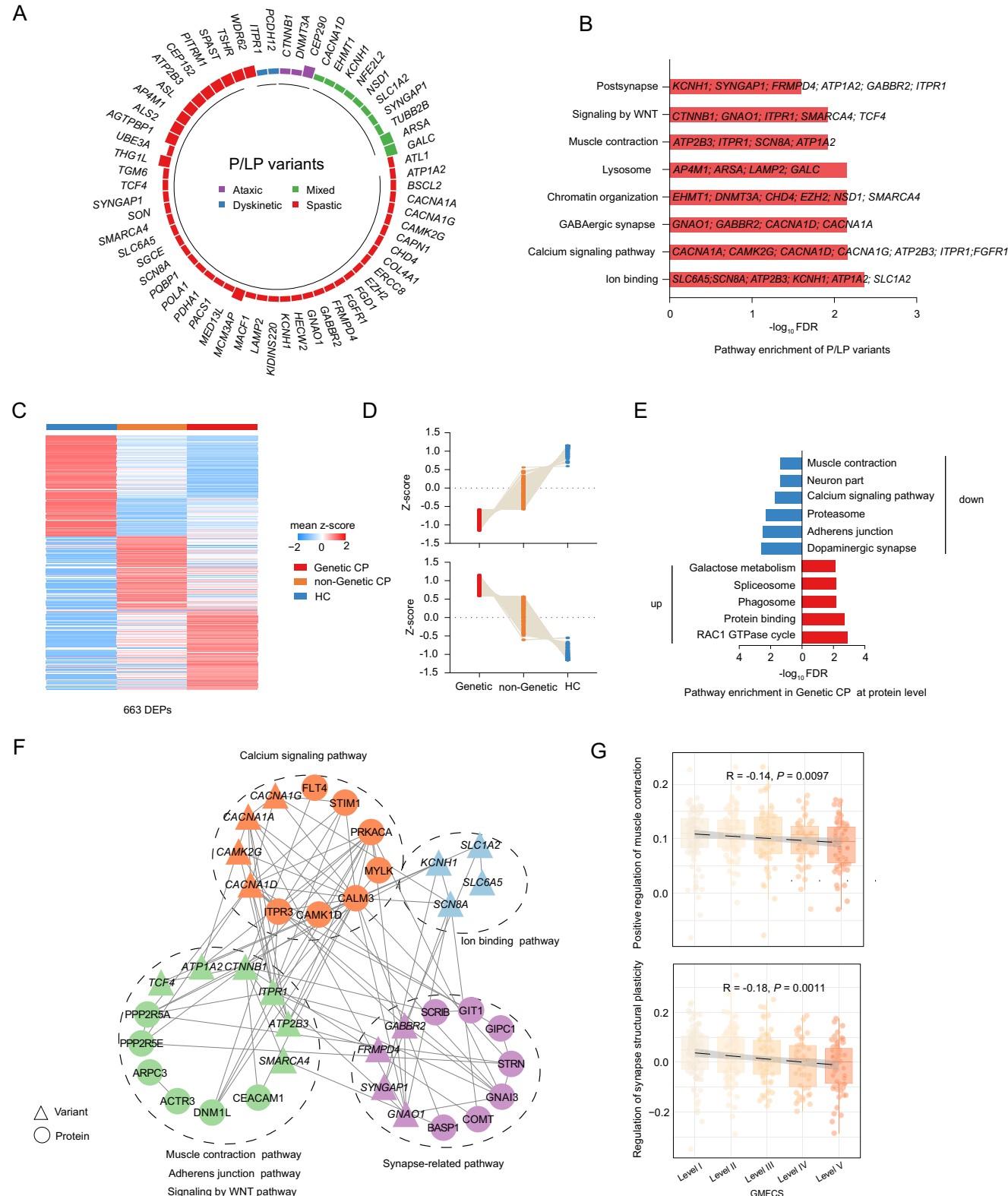

muscle contraction, and adherens junction pathways (Fig. 3C). In the perinatal subtype, 17P/LP variants were identified, including *CAMK2G*, *PDHA1*, *SYNGAP1*, *FRMPD4* and *KCNH1*, with significant enrichment in glucagon signaling and postsynaptic pathways. A total of 7 variants were found in the postnatal group, and 16 in the unknown group, which were primarily associated with synaptic signaling pathways.

To further analyze the biological characteristics of different risk subtypes, we performed ssGSEA analysis at the protein level. The preconception subtype showed upregulation of fatty acid beta oxidation pathway and long-term depression pathway (Fig. 3D). The prenatal subtype was enriched in actin cytoskeleton reorganization and Rhoh GTPase pathways at the protein level (Fig. 3D). The perinatal subtype showed upregulation of the ribosome pathway (Fig. 3D). The postnatal subtype exhibited increased expression in proteins involved in the chronic inflammatory response and calcium dependent phospholipid binding pathways (Fig. 3D). The unknown

**Fig. 2 | Overview of P/LP variants identified in CP. A** The frequency of genetic variants identified in clinical subtypes. **B** Bar chart showing enriched pathways of 76 variants (FDR < 0.05). **C** Heatmap showing proteins that differed between variant CP ($n = 65$), no-variant CP ($n = 256$), and HC ($n = 190$), assessed using the Kruskal-Wallis test for multiple comparisons (FDR < 0.05). **D** The line plots showing upregulated and downregulated serum proteins in genetic CP (Kruskal–Wallis test, FDR < 0.05). **E** Bar chart showing enriched pathways of significantly different proteins detected in the Genetic CP at the protein level (FDR < 0.05). **F** The Variant-protein interaction networks constructed from variants and proteins were significantly altered in the variant CP. Variants and proteins were color-coded according to different pathways. The circles represent proteins and triangles represent variants. **G** Scatterplot showing the association between GMFCS and pathway ssGSEA score (Spearman's correlation test). $P$ value was derived by the two-sided test. The error band represents the 95% CI of the regression line. Sample distribution: GMFCS level I ($n = 90$), level II ($n = 84$), level III ($n = 60$), level IV ($n = 47$), and level V ($n = 63$). Boxplots show median (central line), upper and lower quartiles (box limits), 1.5× interquartile range (whiskers). Source data are provided as a Source Data file.

subtype showed upregulation of the neutrophil activation pathway (Fig. 3D).

Interestingly, we observed downregulation of the beta-catenin/TCF complex assembly pathway, adherens junction pathway, and postsynapse organization pathway in the prenatal subtype, aligning with variant-enriched pathways (Fig. 3D, E). Several adhesion proteins, including SPARC, DSG2, and NRCAM, were significantly downregulated and found to interact with *FGFR1* and *CTNNB1* genes. These findings suggested that genetic variants associated with WNT signaling and adherens junction pathways in prenatal subtype might contribute to downstream pathway alterations at the protein level.

## Proteomic analysis uncovers low birth weight and preterm delivery linked with lipid metabolism

To further explore the association between risk factors and clinical subtypes, we conducted an odds ratio analysis to assess how various risk factors influence biological alterations across CP subtypes (Fig. 4A, B). Notably, PVL, low birth weight (LBW), and preterm birth were significantly enriched in the spastic subtype, whereas the dyskinetic subtype showed stronger associations with pathological jaundice. Among these, LBW and preterm birth demonstrated the highest enrichment, indicating a strong correlation with clinical subtypes ($p < 0.05$) (Fig. 4C).

Given the high significance, we further explored the impact of LBW and preterm birth in the spastic subtype. Initially, we stratified patients into two distinct categories based on the severity of their motor conditions, as delineated by a previous study[32]. Children with more profound motor impairments (levels IV–V) were classified under 'high GMFCS', while those with less severe motor disabilities (levels I to III) were categorized under 'low GMFCS'. Intriguingly, we observed that birth weight (BW) and gestational age (GA) were significantly lower in the high-GMFCS group ($p < 0.05$), indicating that individuals with low birth weight or low gestational age (LGA) exhibited more severe motor deficit (Fig. 4D, E).

To gain a clearer understanding of protein-level changes associated with LBW and LGA, we calculated the correlations between birth weight, gestational age, and protein expression in the spastic subtype (Fig. 4F). We found that proteins associated with LBW or LGA were significantly enriched in lipid metabolism (e.g., APOC2, APOH, and PLTP), oxidoreductase complex (e.g., NDUFA10, CYB5R3, PRDX6, and PRDX5), and antigen binding (e.g., HLA-H and HLA-A) (Fig. 4G, H). In contrast, proteins associated with high birth weight or high gestational age showed significant enrichment in signaling pathways such as cellular detoxification (AKR7L and AKR7A3) and gap junction (e.g., TUBA1A and TUBB1) (Fig. 4G, H). The ssGSEA further showed that lipid-related signaling pathways were significantly negatively correlated with body weight and gestational age (Supplementary Fig. 2A), indicating that LBW/LGA cases exhibit distinct metabolic signatures.

Among lipid metabolism proteins, APOC2 emerged as a key molecule negatively correlated with birth weight and positively associated with GMFCS severity ($r = 0.19$, $p = 0.0017$) (Fig. 4I). Proteins positively correlated with APOC2 were significantly enriched in lipid metabolism (e.g., APOE, APOB) and cytokine production pathway (e.g., IFI16 and MIF) (Fig. 4J). In addition, lipid metabolism pathways showed significant positive correlations with monocyte-related inflammatory pathways (Fig. 4K), suggesting that lipid metabolism disorders might influence immune responses in spastic CP. Furthermore, both lipid metabolism and inflammation-associated proteins were positively correlated with GMFCS scores and negatively associated with birth weight and gestational age (Fig. 4L and Supplementary Fig. 2B). These findings suggested that spastic CP cases with LBW or LGA might lead to dysregulation of lipid metabolism and immune pathways, which are both associated with more severe motor dysfunction (Fig. 4M).

## The serum protein co-expression network unveils associations between immunoglobulins and the GMFCS levels in CP

To investigate the association between serum proteomic profiles and clinical characteristics of CP, we performed WGCNA analysis, which grouped the proteins into seven strongly co-expressed module eigengenes (MEs) at the protein level: MEbrown, MEgreen, MEblue, MEyellow, MEturquoise, MEred, and MEgrey[33] (Fig. 5A, Supplementary Data 1). Distinct modules exhibited significant correlations with clinical indicators. Specifically, the MEblue module was significantly associated with common complications of CP, including epilepsy, as well as hearing and vision impairments. The MEturquoise module was enriched in patients with intellectual disability. The MEred module showed a negative association with the ataxic subtype, while both MEred and MEgreen were enriched in the dyskinetic subtype. Furthermore, both the MEgreen and MEbrown modules were positively correlated with GMFCS levels (Fig. 5A and Supplementary Fig. 3A).

Pathway enrichment analysis unveiled the diverse biological functions of proteins in different module proteins (Fig. 5B). MEblue was enriched in lipoprotein metabolism and complement cascade pathways (Fig. 5B and Supplementary Fig. 3B). MEyellow was characterized by actin cytoskeleton-related pathways, while MEred was enriched in cellular response to heat stress and Rho GTPase signaling pathways (Fig. 5B and Supplementary Fig. 3B). MEgreen and MEturquoise were co-expressed with immunoglobulin gene families IGHV, IGKV, and IGLV, respectively (Fig. 5B). Noting that distinct modules were enriched for specific immunoglobulins, we performed ssGSEA scoring for IGHV, IGLV, and IGKV at the protein level ("Methods"). Comparison between CP and health control serum samples showed a significant upregulation of IGHV-associated immunoglobulins in CP ($p < 0.05$) (Fig. 5C). Moreover, IGHV ssGSEA score also exhibited a significant positive correlation with GMFCS levels ($r = 0.26$, $p = 2e{-}6$) (Fig. 5D; "Methods").

Given the positive correlation between IGHV scores and GMFCS levels, we further examined the impact of the IGHV score on disease pathology. Pathway enrichment analysis showed that proteins positively correlated with IGHV score were mainly enriched in cytokine pathway (e.g., HMGB1, MIF), cholesterol metabolism pathway (e.g., APOE, APOB, and APOC2) and proteasome degradation pathway (e.g., PSMB10, PSME2, and PSMD3), whereas negatively correlated proteins were mainly associated with phagosome pathway (e.g., RAB7A and EEA1), glutathione metabolism pathway (e.g., GPX1 and GPX3) and tight junction signaling (e.g., CTTN and ACTN4) (Fig. 5E, F). These results highlighted the involvement of IGHV score-associated proteins in key biological processes, including inflammation, lipid metabolism, and cellular structural integrity, potentially influencing CP pathology.

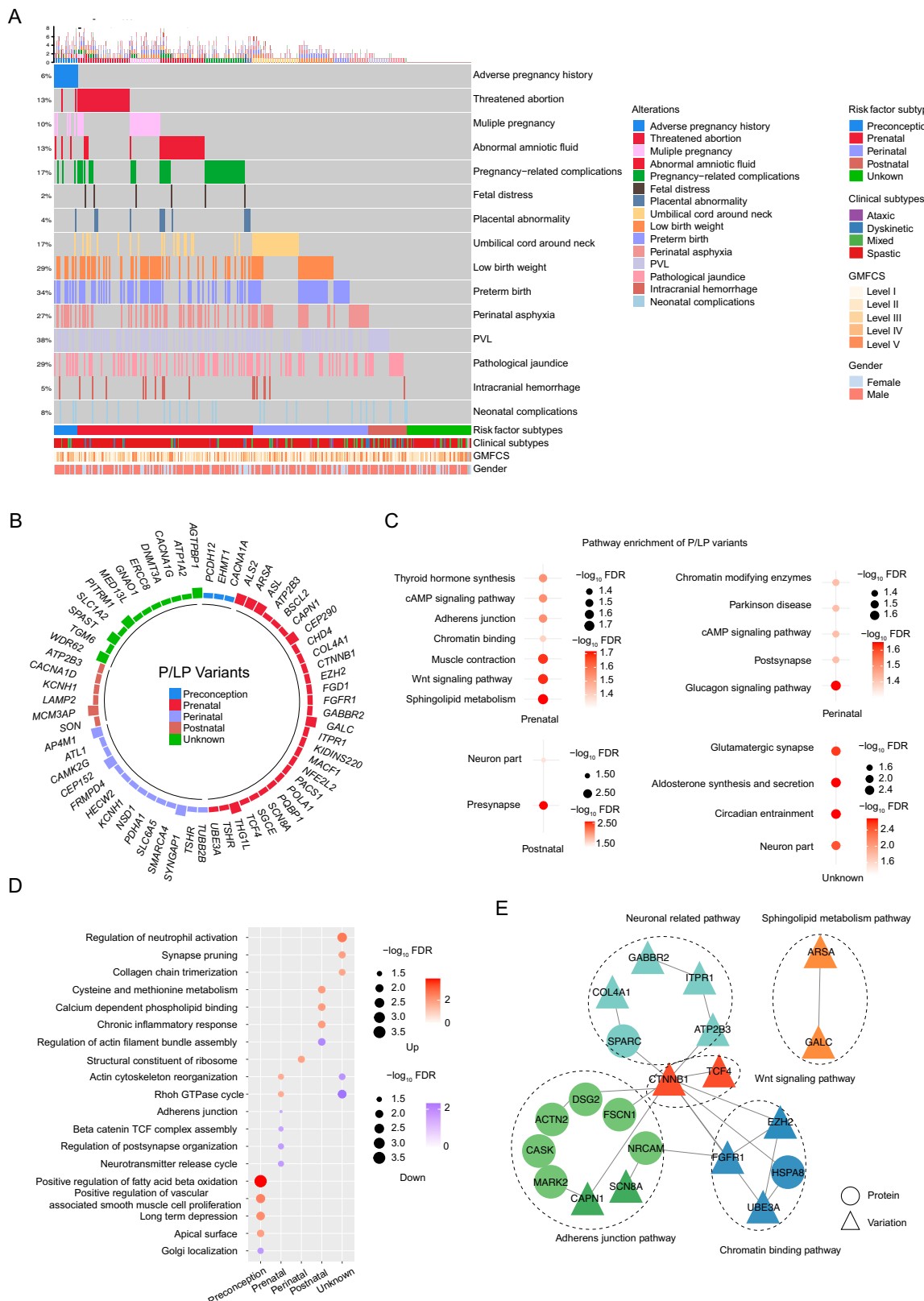

Finally, we developed a predictive model to improve the classification of GMFCS levels based on serum protein profiles. CP samples were randomly divided into training and validation sets at a 4:1 ratio. Protein feature importance was assessed using the XGBoost algorithm with 10-fold cross-validation ("Methods", Fig. 5G). The top 10 proteins were selected to construct a multi-marker predictive model, which

achieved an AUC of 0.76, with an accuracy of 0.75, weighted precision of 0.76, weighted recall of 0.75, and weighted F1-score of 0.75 (Fig. 5H).

In conclusion, we uncovered links between serum protein co-expression networks, particularly immunoglobulins, and GMFCS levels in CP. Using weighted gene co-expression network analysis (WGCNA), we identified six module eigengenes (MEs) correlating with various CP

**Fig. 3 | Characteristics of variants and serum proteins associated with various risk factor types. A** Heatmap illustrating risk factors chronologically across various types of CP. **B** The frequency of genetic variants identified in risk factor subtypes. **C** Bubble plot illustrating distinct enriched pathways of P/LP genetic variants in different risk subtypes (FDR < 0.05). The bubble size corresponds to the significance of the pathways enriched in each group. **D** Bubble plot showing pathways enriched in different risk factor subtypes at the protein level (FDR < 0.05) (two-sided Wilcoxon rank-sum test: one risk factor subtype vs others). The bubble size corresponds to the significance of the pathways enriched in each group. The red bubbles indicate upregulation and the blue bubbles indicate downregulation. **E** The variant-protein interaction networks constructed from variants and proteins significantly altered in the prenatal subtype. Variants and proteins were color-coded according to different pathways. The circles represent proteins and triangles represent variants. Source data are provided as a Source Data file.

clinical features. Notably, the correlation between GMFCS levels and specific IGHV-immunoglobulins highlights their potential as biomarkers for CP severity.

## Discussion

CP encompasses a heterogeneous group of non-progressive neurodevelopmental disorders. Given the complexity of CP's risk factors, identifying serum biomarkers could enable earlier detection of CP, thereby allowing for the initiation of intervention therapies sooner than currently possible[5]. To achieve this, we performed a proteomic analysis to identify serum protein alterations associated with CP, utilizing machine learning to create a multi-biomarker panel for its indication.

Compared with single biomarkers, multi-marker panels offer notable advantages in improving diagnostic accuracy, robustness, and generalizability[34–36]. In this study, we identified ten candidate biomarkers through MS screening and validated them using ELISA in an independent cohort, ultimately establishing a multi-marker model with high predictive accuracy. While mass spectrometry remains a powerful discovery platform, its high cost and technical demands limit its practical use in clinical settings. To facilitate broader implementation, future efforts may benefit from the development of high-affinity ELISA antibodies targeting CP-related proteins, which could enhance sensitivity, specificity, and batch-to-batch consistency for application in routine diagnostics.

Nevertheless, translating biomarker research into clinical practice remains challenging. A key step toward clinical utility is the development of standardized and scalable biomarker integration strategies. To this end, we constructed a logistic regression model using ELISA-derived protein expression levels and established a quantitative risk scoring system that effectively distinguished CP patients from HC individuals. Importantly, the application of the Youden index enabled the definition of an optimized diagnostic threshold, enhancing both interpretability and usability of the model in potential clinical workflows[37,38]. In addition, we identified an optimized three-protein panel (SPARC, DHX9, and MANBA) that retained good diagnostic performance while significantly reducing assay complexity. This suggested that a cost-effective, streamlined implementation of multi-marker strategies may be feasible in routine clinical practice. The modeling and simplification framework might serve as a useful reference for biomarker development, CP. Future multicenter studies will be essential to validate the clinical utility of this diagnostic strategy.

In addition to constructing a diagnostic classifier, we next investigated how rare genetic variants might be associated with altered protein expression profiles. Our previous study revealed that the genetic variants were predominantly enriched in synaptic signaling, WNT and chromosomal modification signaling pathways[17]. However, there is limited research on how these variants relate to CP at the protein level. In this study, our integrative genomic and proteomic analysis demonstrated that patients with genetic CP not only carry rare damaging variants but also exhibit altered expression of key proteins involved in neuronal signaling. For instance, several synapse-related signaling proteins, including GIT1, BASP1, and SCRIB, were found to be downregulated in genetic CP. Notably, GIT1 is an ARF GTPase-activating protein (ARF-GAP) that regulates postsynaptic structure and signaling, and is essential for synaptic plasticity[39]. In addition, we observed reduced expression of calcium-related signaling proteins,

including CALM3, ITPR3, and CAMK1D, suggesting potential disruption of calcium-mediated neuronal signaling in CP. These findings indicated that rare damaging variants might contribute to CP pathogenesis by perturbing synaptic organization and calcium homeostasis.

In addition to genetic factors, clinical risk factors such as low birth weight and preterm birth are well-established contributors to CP[11,40]. Consistent with previous reports[16,41], LBW and preterm birth were significantly enriched in spastic CP in our cohort. In addition, we also found that patients with low birth weight and low gestational age tended to have higher GMFCS levels[42]. Further analysis revealed that spastic CP patients with LBW and LGA exhibited dysregulated lipid metabolism and enhanced inflammatory responses. Cholesterol accumulation affects the vascular system and compromises the blood-brain barrier, leading to or exacerbating neurological disorders[43–45]. Among the molecules altered in lipid metabolism, APOC2 was also associated with higher GMFCS levels. Despite limited reports linking APOC2 to CP, our further analysis revealed that APOC2 is associated with pro-inflammatory cytokines, potentially contributing to the inflammatory processes. This finding supports the hypothesis that lipid metabolism disruptions play a critical role in CP-related inflammation[46], warranting functional investigation of APOC2 in future studies.

Given that motor dysfunction is a core clinical manifestation of CP, we further explored whether serum proteomic profiles could reflect disease severity as measured by GMFCS levels. Though its association with immune function has been suggested, the underlying molecular mechanisms have not been fully delineated[47]. Our research revealed a correlation between serum protein alterations and CP severity, notably IGHV-Immunoglobulins, suggesting a potential immunological link with CP severity. IGHV-related proteins have been reported to be significantly upregulated in the plasma of patients with ASD[48]. Immunoglobulin abnormalities might be related to the pathogenesis of brain injury or might be a complication that affects the severity of the disease, consistent with the persistent inflammation previously found in CP[49]. Additionally, unlike previous studies that relied on clinical indicators for GMFCS severity prediction[50], our study developed a serum proteome-based predictive model.

In summary, this study not only established a robust diagnostic classifier for CP but also revealed protein expression abnormalities associated with genetic and clinical risk factors, offering insights into disease mechanisms (Fig. 5I): (1) The development of a diagnostic classifier for CP, (2) Alterations in ionic signaling and synaptic signaling at the protein level in genetic CP, (3) The metabolic disturbance in CP children with low birth weight and preterm birth, and (4) Serum proteomics may have potential in assessing CP severity, particularly as IGHV-related immunoglobulins were positively correlated with GMFCS level. Nevertheless, further validation in larger and more diverse cohorts, as well as mechanistic studies in experimental models, is warranted to confirm the clinical applicability and biological significance of these findings.

## Methods

### Participant recruitment and clinical data collection

The clinical cohorts of Chinese children diagnosed with CP from two medical centers: the Children's Hospital and the Third Affiliated Hospital of Zhengzhou University. It included two groups: a discovery cohort (346 CP children and 190 healthy controls) and an independent

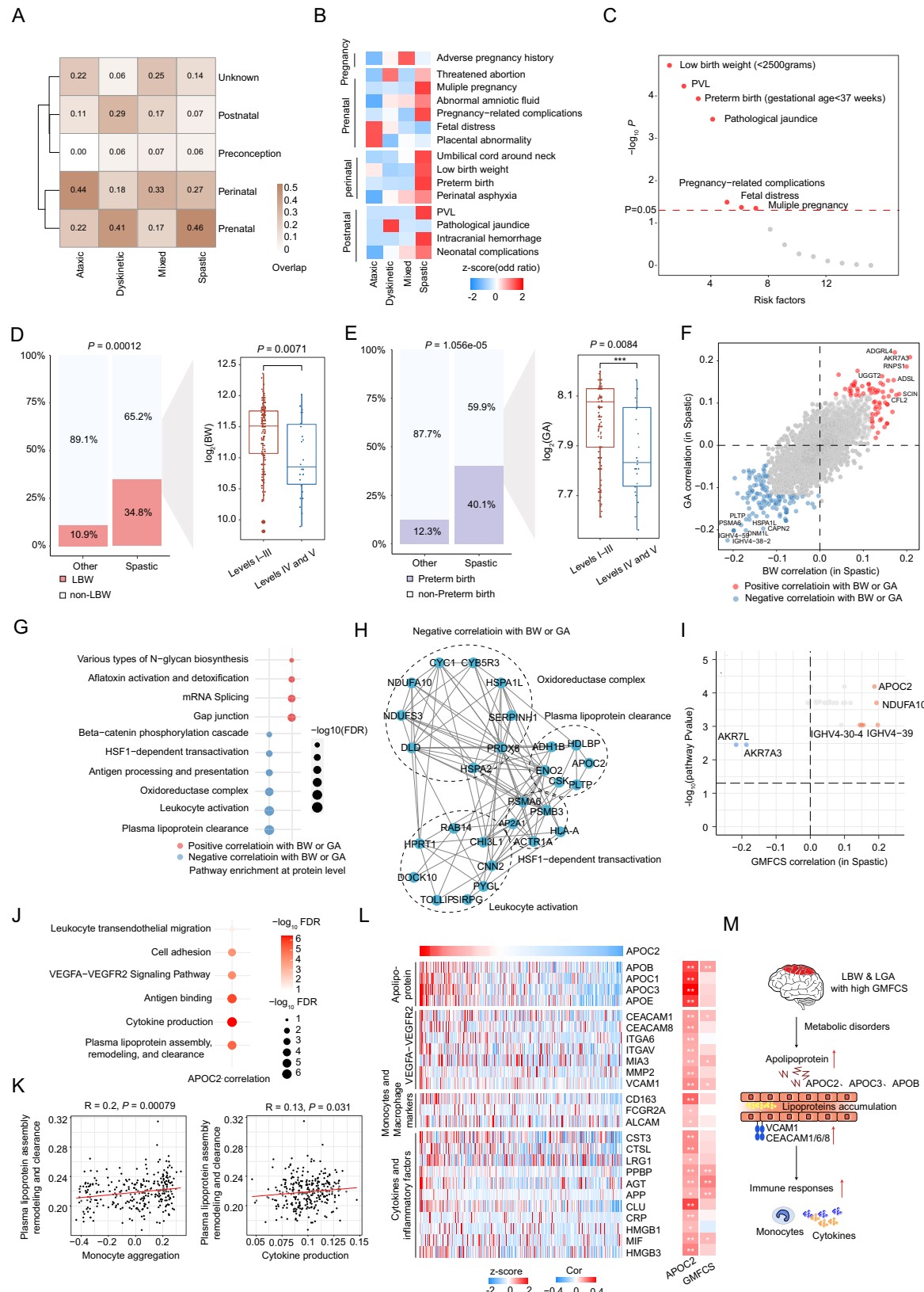

validation cohort (38 CP children and 32 healthy controls). All children diagnosed with CP met Surveillance of Cerebral Palsy in Europe (SCPE) criteria[51], which was consistent with our previous study[16]. The inclusion criteria: (1) All patients were under 18 years of age. For those who received an initial diagnosis before the age of 2, longitudinal follow-up was performed until they reached at least 2 years of age to confirm the

stability and accuracy of the diagnosis. (2) Diagnosed with CP following comprehensive clinical evaluations conducted by pediatric neurologists, including pregnancy and birth history, family history, and detailed physical and neurological examinations. (3) No known chromosomal abnormalities or syndromic diagnoses. The exclusion criteria: (1) Unstable vital signs or serious systemic illness; (2) CP

**Fig. 4 | Molecular characterization of CP children with low birth weight and preterm birth. A** Heatmap showing the relationship between clinical subtypes (columns) and risk types (rows), with each column adding up to one. **B** Heatmap showing the odds ratio for the associations between risk factors and clinical subtypes. **C** The ranked *p*-value from Fisher's exact test between clinical information and risk factors. *P* value was derived by the two-sided test. **D** Bar plot showing the prevalence of low birth weight in the spastic subtype versus other subtypes. The boxplot on the right side shows the birth weight across different GMFCS levels in the spastic subtype (two-sided Wilcoxon rank-sum test). Sample distribution: GMFCS level I–III (*n* = 215) and level IV–V (*n* = 63). **E** Bar plot for preterm birth in the spastic and other subtypes, and the boxplot showing gestational age at different GMFCS levels in the spastic subtype (two-sided Wilcoxon rank-sum test). Sample distribution: GMFCS level I–III (*n* = 215) and level IV–V (*n* = 63). **F** Scatterplot showing the correlation between proteins and two key factors: birth weight and gestational age. **G** Bar plot showing pathways enriched in proteins that were either positively or negatively associated with birth weight or gestational age (FDR < 0.05). **H** The protein–protein interaction networks formed by proteins positively associated with low birth weight or low gestational age. **I** Scatterplot showing the correlation between proteins involved in the pathway, as shown in Fig. 4G and GMFCS (FDR < 0.05). **J** Bubble plot showing pathways enriched in proteins positively associated with APOC2 protein (FDR < 0.05). **K** Scatterplot showing the correlation between scores of serum lipoprotein assembly, monocyte aggregation and cytokine production pathways. **L** Heatmap showing proteins involved in the pathway from Fig. 4J, showing a positive correlation with APOC2. **M** Schematic diagram summarizing the correlation between metabolic disorders and LBW, preterm birth and high GMFCS. For Fig. 4D, E, boxplots show median (central line), upper and lower quartiles (box limits), 1.5× interquartile range (whiskers). For Fig. 4F, I, K, L, Spearman's correlation test was performed, and two-sided *p* values were calculated. Source data are provided as a Source Data file.

secondary to acquired causes (e.g., encephalitis and uncontrolled epilepsy); and (3) CP subtypes predominantly characterized by hypotonia or rigidity. All patients were classified into clinical subtypes of CP according to ICD-10 codes[52]. The diagnostic classification and clinical information used in this study were determined after the age of 2 years for these early-diagnosed cases.

Healthy control samples were obtained during routine pediatric medical examinations at the Third Affiliated Hospital of Zhengzhou University. The diagnostic criteria were consistent with our previous studies[53,54]. The inclusion criteria: (1) Typically developing children with no history of neurological, genetic, or developmental disorders. (2) No recent perinatal complications or current medications. The exclusion criteria: (1) Recent infections, fever, or vaccination within two weeks; (2) First-degree family history of neurodevelopmental or inherited genetic disorders. Additional clinical information is provided in Supplementary Data 1.

### Ethical approval
Informed consent was obtained from all participants and their guardians, with a thorough understanding of the study's purpose and consent for the publication of the results. The research received approval from the Ethics Committee of The Third Affiliated Hospital of Zhengzhou University (Ethical Approval No. 2017-09) and complied with relevant guidelines and regulations. Healthy controls were recruited from children experiencing kindergarten, ensuring a representative sample of the general population.

### Serum sample collection
Peripheral venous blood (1–2 mL per participant) was collected using standard yellow-top vacutainer tubes containing GEL and clot activator (Shandong Acealife Medical Devices Co., Ltd., China). Whole blood samples were allowed to clot at room temperature and subsequently centrifuged at 1000×*g* for 10 min at 4 °C to separate the serum. All samples were processed within 2 h of collection. The resulting serum was aliquoted into 2.0 mL sterile cryovials (Corning®, Cat. No. 430659) and stored at −80 °C until further analysis.

### Whole-exome sequencing data
The variant results for 321 patients among 346 CP were obtained from our previous study[17]. Briefly, variants were annotated using ANNOVAR (v2020-06-08). Variants were retained if they were located in genes listed in the OMIM and HGMD databases and associated with CP phenotypes. These variants were subsequently categorized as pathogenic (P), likely pathogenic (LP), of uncertain significance (VUS), benign (B), or likely benign (LB) according to ACMG/AMP guidelines. Because pathogenic and likely pathogenic (P/LP) variants are considered clinically significant and linked to disease susceptibility[26], all such variants identified in the study were confirmed by Sanger sequencing.

Individuals were classified as genetic CP if they carried at least one P/LP variant in a gene with a known inheritance pattern and a clinical phenotype consistent with the gene's associated disorder. Patients without any such variants were classified as non-genetic CP. More details for these variants are listed in Supplementary Data 1.

### Cell line
The HEK293T cell line (ATCC CRL-11268, RRID: CVCL_QW54) used for quality control was obtained from the Chinese Academy of Sciences and cultured in Dulbecco's Modified Eagle Medium (DMEM; Gibco), supplemented with 10% fetal bovine serum (FBS; Gibco) and 1% penicillin-streptomycin (Sigma-Aldrich). Cells were maintained at 37 °C in a humidified atmosphere with 5% $CO_2$. All cell lines underwent routine testing for mycoplasma contamination and were authenticated using short tandem repeat profiling.

### LC−MS/MS analysis
**Serum protein extraction and trypsin digestion.** Consistent with previous researches[19,20], 2 μL of serum samples were firstly mixed with 98 μL 50 mM NH4HCO3 buffer and proteins were inactivated at 95 °C for 3 min. After cooling to room temperature, the solution was digested with trypsin at a 1:25 enzyme-to-protein ratio at 37 °C for 16 h. The digestion was halted by adding 10 μL of ammonia solution. The peptides were then dried using a SpeedVac at 60 °C. Next, 100 μL of 0.1% formic acid (FA) was added to dissolve the peptides. This was followed by 3 min of vortexing and centrifugation at 1000×*g* for 2 min. For desalination, the 3 M C18 membrane pillars were activated using 100 μL of 100% acetonitrile and 50% acetonitrile, each twice, and balanced with 100 μL of 0.1% FA. The peptides were loaded onto the activated C18 membrane and eluted twice with 100 μL of 0.1% formic acid (FA). Finally, 100 μL of elution buffer (0.1% FA in 50% acetonitrile) was added to the pillars for elution twice, and only the effluent was collected. The collected peptides were then dried using a vacuum drier at 60 °C.

**LC–MS/MS detection.** Consistent with a previous study[20], peptides were analyzed using a Q Exactive HF-X Hybrid Quadrupole-Orbitrap Mass Spectrometer (Thermo Fisher Scientific) in conjunction with a high-performance liquid chromatography system (EASY nLC1200, Thermo Fisher Scientific). Dried peptide samples were re-dissolved in 100 μL of Solvent A (0.1% formic acid in water). The peptide concentration for each sample was measured using a NanoDrop (Thermo Fisher Scientific) at 280 nm absorbance. The standard loading of 200 ng peptide was then loaded onto a 75 μm-inner-diameter column with a length of 9 cm (1.9 μm ReproSil-Pur C18-AQ beads, Dr. Maisch GmbH) over a 10-min gradient (Solvent A: 0.1% FA in water; Solvent B: 0.1% FA in 80% acetonitrile) at a constant flow rate of 600 nL/min. Eluted peptides were ionized at 2 kV and introduced into the mass spectrometer. Mass spectrometry was conducted in data-independent

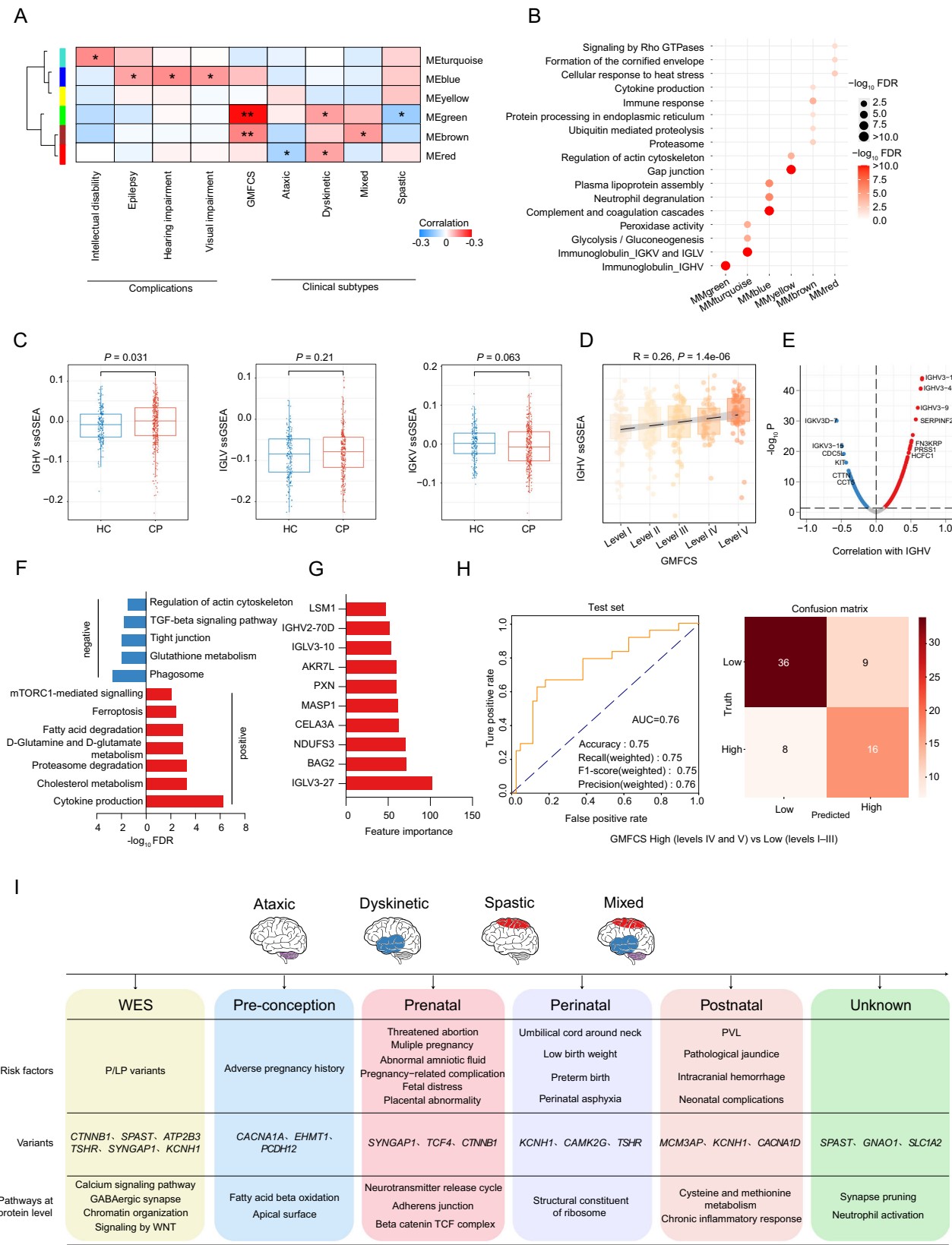

acquisition mode (DIA). For the MS1 spectra, ions ranging from m/z 300 to 1400 were captured by an Orbitrap mass analyzer at a resolution of 30,000. The automatic gain control (AGC) target value was set to 3E6, and the maximum ion injection time was limited to 20 ms. Subsequently, DIA segments were acquired at a resolution of 15,000 with an AGC target of 1E6. The default charge state set for MS2 was 2.

**Peptide identification and protein quantification.** All raw data were processed using "Firmiana" (https://phenomics.fudan.edu.cn/firmiana/)[55]. The data against UniProt human protein database (updated on 2019.12.17, 20406 entries) using FragPipe (v12.1) with MSFragger (2.2)[56]. The mass tolerances were 20 ppm for precursor and 50 mmu for product ions. Up to two missed cleavages were allowed. The

**Fig. 5 | WGCNA reveals an association between the IGHV-related protein module and higher GMFCS levels. A** Heatmap showing the correlation between modules obtained from WGCNA analysis and various clinical outcomes. *$p < 0.05$; **$p < 0.01$; ***$p < 0.001$. **B** Enrichment pathway of different modules at the protein level (FDR < 0.05). The bubble size and color correspond to the significance of the pathways enriched in each model. **C** Boxplots showing the ssGSEA score of IGHV, LGLV, and IGKV between CP and healthy controls at the protein level (two-sided Wilcoxon rank-sum test). Sample distribution: CP ($n = 346$) and HC ($n = 190$). **D** Scatterplot showing the Spearman's correlation between the IGHV score and GMFCS. *P* value was derived by the two-sided test. The error band represents the 95% CI of the regression line. Sample distribution: GMFCS level I ($n = 90$), level II ($n = 84$), level III ($n = 60$), level IV ($n = 47$), and level V ($n = 63$). **E** Volcano diagram showing the Spearman's correlation between the IGHV score and proteins. *P* value was derived by the two-sided test. **F** Bar plot showing pathways enriched in proteins significantly associated with IGHV ssGSEA score at the protein level (FDR < 0.05). **G** Bar plot showing the top 10 proteins for feature importance using the XGBoost-based model. **H** Graphical summary highlighting key molecular findings of integrated analysis of the serum proteomes from 346 CP children and 190 healthy controls. **I** Schematic diagram summarizing. For Fig. 5C, D, boxplots show median (central line), upper and lower quartiles (box limits), 1.5× interquartile range (whiskers). Source data are provided as a Source Data file.

search engine set cysteine carbamidomethylation (C) as a fixed modification and N-acetylation and oxidation of methionine as variable modifications. Precursor ion score charges were limited to +2, +3, and +4. The data were also searched against a decoy database so that protein identifications were accepted at a false discovery rate (FDR) of 1%. The results of DIA data were combined into spectra libraries using SpectraST software. A total of 327 libraries were used as reference spectral libraries, which were used in our previous studies[57,58]. DIA data were analyzed using DIA-NN (v1.7.0)[59]. The default settings were used for DIA-NN (Precursor FDR: 1%, Log lev: 1, Mass accuracy: 20 ppm, MS1 accuracy: 10 ppm, Scan window: 30, Implicit protein group: genes, Quantification strategy: robust LC (high accuracy)). Quantification of identified peptides was calculated as the average of chromatographic fragment ion peak areas across all reference spectra libraries. Label-free protein quantification was performed using an intensity-based approach, incorporating delayed normalization and maximal peptide ratio extraction (MaxLFQ)[60]. The fraction of total (FOT) was used to represent the normalized abundance of a particular protein across samples. It was calculated by dividing the intensity of each protein by the total intensity of all proteins within the same sample[19]. The FOT values were multiplied by $10^5$ for the ease of presentation.

**Quality control of the mass spectrometry data.** Strict quality control protocols were implemented to ensure the reliability of MS performance and the entire experimental process. Specifically, pooled serum samples, mixed by all CP and HC samples in our study, were included to monitor overall technical consistency, while human embryonic kidney-derived HEK293T cell lysates served as external QC standards to evaluate platform stability and performance across runs. The preparation of pooled QC samples followed the same protocol as the cohort serum samples. This resulted in a total of 20 HEK293T QC runs and 11 pooled QC runs. Mass spectrometry assays for both the pooled QC and HEK293T samples were aligned with those of the disease and control samples to ensure consistency.

**Missing value imputation.** Proteins detected in more than 30% of the samples were selected both within the CP and healthy control groups. The missing values were imputed by K-nearest neighbor (KNN) algorithm using the five nearest neighbors based on R package "impute" (version 1.68.0).

**Global proteomics data analysis**
**Differential protein analysis.** Proteins expressed in more than 30% of the samples were selected for differential expression analysis. The two-sided Wilcoxon rank-sum test was employed to identify proteins with significantly different expression between CP and HC samples. Differentially expressed proteins were classified as up-regulated or down-regulated based on an FDR < 0.05 and a fold change threshold (CP vs HC: >1.5 or <0.67). Additionally, the Kruskal–Wallis test was applied to assess significant differences in protein expression across three groups: genetic CP, non-genetic CP, and healthy controls (FDR < 0.05).

**Pathway enrichment analysis.** Differentially expressed proteins and P/LP genes were subjected to pathway enrichment analysis ConsensusPathwayDB[61] (http://cpdb.molgen.mpg.de/). We identified significantly enriched pathways based on threshold of FDR < 0.05.

**Single-sample gene set enrichment analysis.** We performed ssGSEA using the R package GSVA (version 1.42.0)[62,63]. Proteins expressed in more than 30% of the samples were used for subsequent analysis. Gene sets from databases such as KEGG, Reactome, Gene Ontology, and HALLMARK, sourced from the MsigDB database (v7.4), were used as the reference background for this analysis.

**Immunoglobulin score.** The IGHV score, IGLV score, IGKV score and IGLC score were computed using ssGSEA[62] via the GSVA package[63] based on 124 human immunoglobulins as background genes in our proteomics dataset, including IGHV-related, IGLV-related, IGKV-related and IGLC-related proteins.

**Weighted gene co-expression network analysis (WGCNA).** We used the R package WGCNA (1.72.1) to organize proteins into functional models[33]. Proteins with less than 50% missing values were subjected to analysis. Standard parameters were changed to a soft threshold at power of 2 (based on scale free topology model fit, $R^2 = 0.85$), a "unsigned" network, average clustering, and a minimum module size of 20. The algorithm assigned proteins into 7 distinct modules determined by the Pearson correlation coefficients between module eigengenes and specific traits. For the modules identified, pathway enrichment analysis was used for functional annotation. The correlation between genes and their corresponding modules was represented by the module membership (MM). The correlation between genes and clinical information was represented by the gene significance (GS).

**Protein–protein interaction annotation.** We used the STRING database (12.0) (https://cn.string-db.org) to explore the protein-protein interactions and pathway enrichment.

**Construction and validation of predictive models.** To diagnose the patient from healthy controls, we randomly split the 536 samples into training and testing sets (at a ratio of 4:1), and only proteins with less than 50% missing values across samples were retained. Subsequently, protein expression values were log2-transformed across all samples. We used the Python XGBoost (version 2.1.0) with 10-fold cross-validation to develop a predictive model. In the first step, to reduce redundancy and improve model stability and interpretability, highly correlated proteins were removed, retaining only the most representative features with strong associations to disease status. In the second step, statistical testing was conducted on the remaining proteins using the limma package (version 3.50.3) in R. Proteins with a false discovery rate (FDR) < 0.05 were considered significant, and sex and age were additionally included as covariates in the model. In the final step, we applied an XGBoost-based feature ranking algorithm to assess the discriminative importance of these DEPs. Hyperparameter tuning was performed using GridSearchCV, and the final classification model was constructed using XGBClassifier.

Moreover, the expression values of each biomarker in both the CP and HC groups were winsorized to the range of mean ± 3 standard deviations to mitigate the influence of outliers. This approach effectively reduces the impact of extreme values while preserving the overall distribution of the data, and has been widely adopted in previously published studies[64,65]. The independent cohort was randomly divided into training and testing sets at a 6:4 ratio for machine learning analysis.

In order to assess the severity of the disease, we randomly divided the 344 CP samples into training and test sets in a 4:1 ratio. We used the Python XGBoost (version 2.1.0) with 10-fold cross-validation to develop a predictive model. In the training set, we selected the top 10 important features from significant proteins. We applied RandomizedSearchCV with 100 iterations to optimize the hyperparameters of the XGBoost classifier. All protein expression values were log2-transformed across all samples. The final test model was trained using XGBoost: XGBClassifier with the following parameters: scale_pos_weight=2.

**Risk score and optimal cut-off value.** A logistic regression model was constructed using the ELISA expression levels of the 10 candidate biomarkers to calculate a composite risk score for each participant, implemented in Python scikit-learn (version 1.5.0). The optimal diagnostic threshold was determined by maximizing the Youden Index (sensitivity + specificity − 1) on the ROC curve. This strategy is consistent with widely adopted approaches in the literature and offers good generalizability and reproducibility[37,38]. The composite risk score was calculated using the following formula:

$$\text{Risk score} = W_0 + W_1 \times \text{Protein}_1 + W_2 \times \text{Protein}_2 + \ldots + W_n \times \text{Protein}_n \quad (1)$$

In this formula, $W_0$ represents the intercept, while $W_1$, $W_2$, ... $W_n$ are the weighted coefficients corresponding to each biomarker. $\text{Protein}_1$, $\text{Protein}_2$, ... $\text{Protein}_n$ are the log-transformed ELISA-derived expression values of the 10 selected biomarkers.

**Enzyme-linked immunosorbent assay (ELISA).** Quantification of target proteins was performed using enzyme-linked immunosorbent assay (ELISA), following the manufacturers' protocols for each kit. Absorbance was measured using the TECAN SPARK 10 M microplate reader (Tecan, Switzerland) to ensure high-precision optical detection.

A panel of ELISA kits targeting specific biomarkers was employed, sourced from validated commercial providers. The proteins measured and their corresponding kit sources and identifiers were as follows: ARHGEF10 (Animalunion Biotechnology Co., Cat# LV11301), BCAR1 (Finetest Co., Cat# EH2694), CUTA (Animalunion Biotechnology Co., Cat# LV11306), LONP1 (Finetest Co., Cat# EH9823), SPARC (Elabscience, Cat# E-EL-H1351), ANXA2 (Yuanju Biotechnology Co., Cat# YJ326723), MANBA (Yuanju Biotechnology Co., Cat# YJ580091), MME (Yuanju Biotechnology Co., Cat# YJ105983), GNAI3 (Yuanju Biotechnology Co., Cat# YJ360818), and DHX9 (Animalunion Biotechnology Co., Cat# LV11303).

**Statistical analysis.** The statistical analysis was performed according to previously established methods[22]. In brief, clinical data were analyzed using standard statistical methods in R (version 4.1.2) and Python (version 3.10) environments, including Fisher's exact test, Wilcoxon rank-sum test, and Kruskal-Wallis test. Fisher's exact test was used for categorical variables, while the Wilcoxon rank-sum and Kruskal–Wallis tests were applied for assessing differences between categorical and continuous variables across subgroups. Additionally, Spearman's correlation was used to evaluate relationships between continuous variables.

**Reporting summary**
Further information on research design is available in the Nature Portfolio Reporting Summary linked to this article.

## Data availability
The raw mass spectrometry proteomic data generated in this study have been deposited to the ProteomeXchange Consortium (https://proteomecentral.proteomexchange.org) via the iProX partner repository[66,67] with the dataset identifier PXD068404. Additional processed data supporting the findings are available in the Supplementary Information or Source Data files. Source data are provided with this paper.

## Code availability
No custom code or mathematical algorithms were used in this study. All analyses were conducted using established methods and standard software ("Methods"). The code used to generate figures is available from the corresponding authors upon request.

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

## Acknowledgements

This work was supported by the National Key Research and Development Program of China (2022YFA1303200 [C.D.], 2022YFA1303201 [C.D.], and 2022YFC2704801 [C.Z.]). National Natural Science Foundation of China (32330062 [C.D.], 31972933 [C.D.], and U21A20347 [C.Z.]), sponsored by Program of Shanghai Academic/Technology Research Leader (22XD1420100 [C.D.]), the Major Project of Special Development Funds of Zhangjiang National Independent Innovation Demonstration Zone (ZJ2019-ZD-004 [C.D.]), Shanghai Municipal Science and Technology Major Project (2023SHZDZX02 [C.D.]), the Fudan Original Research Personalized Support Project [C.D.], the grant from Department of Science and Technology of Henan Province for international collaboration (GZS2023003 [X.W.], 241111521300 [C.Z.]) and the Health Department of Henan Province (SBGJ202301009 [C.Z.]), Swedish governmental grants to scientists working in health care (ALFGBG-1005209 [C.Z.], ALFGBG-1005257 [X.W.]), the Swedish Research Council (2021-01950 [X.W.], 2022-01019 [C.Z.]). This work is supported by the Shanghai Municipal Science and Technology Major Project, the Human Phenome Data Center of Fudan University, and the Shanghai Phenomic precision measurement professional technical service platform (23DZ2290800).

## Author contributions

C.D. and C.Z.: conceptualization and methodology. Y.X., C.M., C.D., and C.Z.: project administration. C.D., X.W., and C.Z.: funding acquisition. Y.X., Y.S., H.G., L.L.Z., J. Zhang, Y. Wu, X.Z., L.R.Z., D.Z., and X.W.: resources (sample collection, clinical data and follow-up information collection). C.M. and S.H.: investigation (samples handling and proteomic data generation). S. Tan, S. Tian, and J.F.: mass spectrometry data generation and instrument maintenance. Y. Wang and Q.X.: investigation (genomic data generation). C.M., J. Zhu, and S.H.: formal analysis (bioinformatics and statistical analyses). G.H. and C.M.: investigation (ELISA and analyses). C.M. and Y.X.: writing—original draft preparation. M.K., J.G., C.D., and C.Z.: revised the—paper critically for important intellectual content. All authors: writing—review and editing.

## Competing interests

The author declares no competing interests.

## Additional information

[1]Henan Key Laboratory of Child Brain Injury and Henan Pediatric Clinical Research Center, The Third Affiliated Hospital and Institute of Neuroscience of Zhengzhou University, Zhengzhou 450052, China. [2]Clinical Research Center for Cell-based Immunotherapy of Shanghai Pudong Hospital, Fudan University Pudong Medical Center, State Key Laboratory of Genetics and Development of Complex Phenotypes, School of Life Sciences, Human Phenome Institute, Fudan University, Shanghai 200433, China. [3]Department of Human Anatomy, School of Basic Medicine, Zhengzhou University, Zhengzhou, Henan Province, China. [4]Children's Hospital of Fudan University and Institutes of Biomedical Sciences of Fudan University, Shanghai 201102, China. [5]Department of Children Rehabilitation, Third Affiliated Hospital of Zhengzhou University, Zhengzhou, China. [6]Department of Pharmacology, School of Basic Medical Sciences, Academy of Medical Science, Zhengzhou University, Zhengzhou, 450001 Zhengzhou, China. [7]Pediatric Movement Disorders Program, Barrow Neurological Institute, Phoenix Children's Hospital, Phoenix, AZ 85016, USA. [8]Departments of Child Health, Neurology, Genetics and Cellular & Molecular Medicine, University of Arizona College of Medicine Phoenix, Phoenix, AZ 85004, USA. [9]Centre for Perinatal Medicine and Health, Institute of Clinical Sciences, Sahlgrenska Academy, University of Gothenburg, Gothenburg, Sweden. [10]Robinson Research Institute and Adelaide Medical School, The University of Adelaide, Adelaide, SA 5000, Australia. [11]Department of Women's and Children's Health, Karolinska Institutet, 17177 Stockholm, Sweden. [12]Center for Brain Repair and Rehabilitation, Institute of Neuroscience and Physiology, University of Gothenburg, Göteborg 40530, Sweden. [13]These authors contributed equally: Yiran Xu, Chi Ma, Yanyan Sun, Jiajun Zhu, Shiman He. ✉e-mail: Changlian.zhu@neuro.gu.se; chend@fudan.edu.cn

