## [Transparent Peer Review file · Nature Communications]

Serum Proteomics Reveals Diagnostic Biomarkers and Molecular Pathways in Cerebral Palsy

Corresponding Author: Professor Chen Ding

Version 0:

Reviewer comments:

Reviewer #1

(Remarks to the Author)

The study conducted a comprehensive serum proteomics analysis combined with genetic data across large cohorts, leading to the identification of several diagnostic biomarkers for Cerebral Palsy (CP). These findings offer valuable insights for the basic and clinical research of SP. However, there are several significant concerns remain that warrant attention before considering for publication. Here are my questions and comments for the authors to consider:

Major Concerns:

1. Biomarker Identification and Validation

A primary aim of this study is to identify biomarkers for CP. Although XGBoost serves as a robust predictive model, it is sensitive to outliers. The authors provided details on model construction, yet information on data preprocessing, essential for model development, is missing.

2. Furthermore, it is unclear if the Area Under Curve (AUC) reported was derived from the combination of the top 10 validated markers in serum. Clarifying this aspect and disclosing detailed sample characteristics would enhance reproducibility. Additionally, while the accuracy of these biomarkers is critical, it is not presented. Lastly, the protein level variance within groups was greater than that between groups; it would be helpful for the authors to explain how this was addressed.

3. Sample Heterogeneity

Although the discussion mentions overcoming sample heterogeneity, specific details on data processing steps to mitigate this variability would improve clarity and facilitate study replication.

4. Sample Classification

The criteria for classifying samples into genetic CP, non-genetic CP, and healthy controls require clarification. For instance, are patients classified as genetic CP if they harbor at least one pathogenic/likely pathogenic variant?

5. Clarification on Diagnostic Tools (Line 83)

The statement, "While MRI enhances the diagnostic capability for CP, approximately 10% of patients—particularly those with milder forms—exhibit normal MRI results," may be somewhat misleading. In clinical practice, CP diagnosis involves not only MRI but also assessments of brain development and reflex examination. A rephrasing could better reflect the multifaceted diagnostic approach.

6. Age Range in CP Diagnosis (Line 136)

The authors report a wide age range at CP diagnosis, with a mean of 29.6 months (ranging from 2 to 180 months). Did the authors analyze and describe the influence of age on CP diagnosis or progression? This would provide valuable context for interpreting the findings.

Minor Concerns:

The boxplot for EIF3L in Figure 1H is incorrect, as it lacks the sample from the HC group.

(Remarks to the Author)

The strengths of the study are:

Xu and colleagues conducted a proteomic analysis of serum samples from children with cerebral palsy (CP) and healthy controls to identify potential diagnostic biomarkers for CP and to explore the molecular mechanisms underlying the interplay of genetic and environmental risk factors. This study recruited 346 children with CP and 190 healthy children, providing a robust dataset to support the research. The substantial effort invested by the researchers in this study deserves commendation, as it represents a notable strength of their work. The findings of this study are expected to offer valuable insights for the early diagnosis and intervention of CP, with significant clinical implications and practical application value.

The weaknesses of the study are:

Based on a comprehensive reading of the manuscript, the primary issue appears to be a lack of refinement in the logical flow of the narrative. The main body of the text seems to be divided into two parts: the first part focuses on the process of identifying and validating diagnostic biomarkers (up to line 181), while the latter part emphasizes the exploration of the molecular mechanisms linking genetic factors, environmental risk factors, and cerebral palsy. In my view, the connection between these two sections is insufficiently developed in the writing.

Additionally, there are some questions regarding specific details in the manuscript:

Introduction Section

1. The introduction of this paper highlights that MRI is currently an important auxiliary method for diagnosing CP, but approximately 10% of mild CP patients exhibit normal MRI results. Does the CP cohort in this study include such patients? If so, was it possible to validate these biomarkers in this subgroup and achieve results consistent with the current findings? Such validation would further demonstrate the superiority of the proposed diagnostic biomarkers.

Results Section

1. In the second part of the results, differentially expressed proteins (DEPs) between the CP and healthy groups were analyzed, identifying 82 upregulated and 83 downregulated proteins. However, in the machine learning analysis, 24 DEPs were selected as candidate proteins for the training set. What were the criteria for this selection? (lines 171–173)

2. The study identified a panel of 10 candidate protein biomarkers through machine learning and validated them in an independent cohort. However, Figure 1 only presents ELISA results for 9 proteins. What about the results for the remaining protein?

3. According to the results in Figure 1H, while the differences in protein expression are statistically significant, the magnitude of these differences between the CP and control groups appears small. What is your perspective on the clinical implications of such findings for diagnosis, treatment, or prognosis assessment?

4. In the third part of the results (line 207), the study states, “we next classified these samples into genetic CP, non-genetic CP, and healthy controls.” What criteria were used for this classification? Please clarify and provide additional details.

5. The manuscript mentions that low birth weight or preterm birth is closely associated with spastic CP. However, given that spastic CP accounts for a large proportion of the cases collected, could this result in bias in the findings? (line 258, Figure 1B)

6. In Figure 5H, what are the precision and recall corresponding to the reported AUC?

Conclusion Section

1. Based on the results in Figure 1H, how can a diagnostic biomarker combination strategy be formulated? Expanding on this point in the discussion section could provide valuable guidance for future research.

2. In the final conclusion, the statement “The predicted model based on serum proteins could distinguish between different GMFCS levels” seems questionable. Is this claim an overstatement of the study’s findings? In the fifth part of the results, it was only reported that high GMFCS levels are closely associated with specific IGHV-immunoglobulins, but no specific differential proteins were identified across the various GMFCS levels.

Methods Section

1. Regarding the collection of blood samples, how much blood was drawn from each participant? What were the components of the collection tubes? These details are crucial for ensuring the quality of the proteomic analysis.

2. It is well known that obtaining samples from healthy controls is more challenging compared to CP patients. The methods section mentions that the control group samples were collected from healthy children in kindergartens. How were these children recruited? Were there specific inclusion/exclusion criteria for the CP and control groups? The details provided in the manuscript are limited; please supplement this information.

Others

1. In attachment 9836907 sheet1, two samples lack GMFCS classification and other related information. Were these samples excluded from the analysis? Additionally, no data on CP patients is present in sheet7. Furthermore, in attachment 9836908 sheet1, the number of P/LP cases does not match the 76 cases mentioned in the manuscript. Please clarify these inconsistencies.

2. Regarding the manuscript’s writing, there is room for improvement. For example, in line 114, the first occurrence of an English abbreviation should include its full form. There is also an abbreviation error on line 279. Some expressions are ambiguous, such as “Nevertheless, its application in CP remains underexplored, warranting further research to validate its role as a diagnostic tool.” The serum proteomic is a method for discovering diagnostic biomarkers, not a diagnostic tool for the disease itself—diagnostic biomarkers are. The manuscript requires further revisions to enhance its clarity and readability.

3. The references section lacks consistent formatting and requires standardization.

Reviewer comments:

Reviewer #1

(Remarks to the Author)

The authors have fully addressed my concerns. I recommend acceptance.

Reviewer #2

(Remarks to the Author)

Response to Reviewers Point-by-Point

REVIEWER COMMENTS

Reviewer #1 (Remarks to the Author):

The study conducted a comprehensive serum proteomics analysis combined with genetic data across large cohorts, leading to the identification of several diagnostic biomarkers for Cerebral Palsy (CP). These findings offer valuable insights for the basic and clinical research of SP. However, there are several significant concerns remain that warrant attention before considering for publication. Here are my questions and comments for the authors to consider:

Response: We appreciate the reviewer for the positive evaluation and constructive comments. In response to the reviewer’s insightful comments, we have systematically revised the manuscript by: (1) providing clearer and more detailed descriptions of the methods; (2) conducting additional analyses to assess the robustness of the diagnostic performance; and (3) clarifying the classification criteria for genetic and non-genetic forms of cerebral palsy. All changes in the revised manuscript have been marked in red for clarity. The detailed point-by-point responses were provided as follows.

Major Concerns:

R1Q1. Biomarker Identification and Validation

A primary aim of this study is to identify biomarkers for CP. Although XGBoost serves as a robust predictive model, it is sensitive to outliers. The authors provided details on model construction, yet information on data preprocessing, essential for model development, is missing.

Response to R1Q1:

We sincerely apologize for the lack of clarity in the description of the data preprocessing steps in the machine learning pipeline in the original manuscript. **We fully agree with the reviewer that data preprocessing is critical for the development of machine learning models.** Proper data preprocessing not only reduces noise but also significantly enhances the stability and generalizability of predictive models.

In response to the reviewer’s comment, we have added a detailed description of the data preprocessing steps performed prior to machine learning model construction in the revised manuscript (*Cell Res*, 2022, PMID: 36307579; *Cell Rep Med*, 2023, PMID: 37633276). Specifically, we used the **fraction of total (FOT) method to normalize protein-level quantification** data by dividing the intensity of each protein by the total intensity of all proteins within the same sample. This normalization strategy has been widely adopted in proteomic studies to control for sample-level variation (*Journal of Hematology & Oncology*, 2022, PMID: 35659036; *Cell Res*, 2022, PMID: 36307579). Proteins with more than 50% missing values across samples were excluded. Subsequently, **protein expression data were log2-transformed** across all samples prior to machine learning model

construction. These preprocessing steps helped standardize the data and reduce technical variability.

**Moreover, we optimized key XGBoost hyperparameters to improve model robustness and reduce**
**sensitivity to outliers in the original manuscript.** Specifically, we applied a 10-fold cross-validation
strategy and conducted hyperparameter tuning via grid search. The tuning process focused on parameters
that control model complexity, including max_depth and min_child_weight. The final 10-protein panel
was used to build a classification model that exhibited reliable predictive performance on the test set
(AUC = 0.96; accuracy = 0.88; precision = 0.88; recall = 0.91; F1-score = 0.88), supporting both the
reliability of the selected biomarkers and the robustness of the model.

**To further assess the model’s sensitivity to outliers, we applied winsorization in the revised version,**
a widely used approach that reduces the impact of extreme values while preserving the overall data
structure (*Cancer Cell*, 2018, PMID: 29622466; *Nat Aging*, 2021, PMID: 37118015). Specifically,
expression values of 10 biomarker in the CP and HC groups were truncated to the range of mean \pm 3
standard deviations. The classification model was then trained using the outlier-adjusted feature proteins
and evaluated on the testing set based on XGBoost with 10-fold cross-validation. **Notably, the model**
**achieved an AUC of 0.96, with an accuracy of 0.87, precision of 0.88, recall of 0.91, and F1-score of**
**0.90, consistent with the performance observed prior to outlier adjustment (Figure RL1A, see**
**below),** suggesting the stability of the selected biomarker panel.

Thank you again for your professional and constructive comments, which have greatly helped improve
the quality of this manuscript. As our data preprocessing and modeling procedures followed a
standardized and consistent workflow, the classification model showed similar performance before and
after outlier adjustment (AUC = 0.96 in both cases). **To better illustrate the impact of outlier**
**adjustment, we have included the results after outlier adjustment in the updated figures (Figure**
**S1I) of the revised manuscript.**

The related information of data preprocessing was updated in the “**Methods**” section (**lines 586-589 on**
**pages 20; lines 654-656 on pages 22)** and the **outlier adjustment** was updated in the “**Results**” section
(**lines 178-181 on pages 6)** and “**Methods**” section (**lines 667-670 on pages 22-23)** in the revised
manuscript.

Figure RL 1. (A) Receiver Operating Characteristic (ROC) curves evaluating the performance of the 10 combined biomarkers in both the original and outlier-adjusted models, accompanied by confusion matrices illustrating the performance of the machine learning classifier.

**R1Q2.** Furthermore, it is unclear if the Area Under Curve (AUC) reported was derived from the
combination of the top 10 validated markers in serum. Clarifying this aspect and disclosing
detailed sample characteristics would enhance reproducibility. Additionally, while the accuracy of
these biomarkers is critical, it is not presented. Lastly, the protein level variance within groups was
greater than that between groups; it would be helpful for the authors to explain how this was
addressed.

Response to R1Q2:

We appreciate the reviewer’s insightful comments. **In the revised manuscript**, we have provided
detailed clarifications on the model evaluation metrics, cohort characteristics, and the interpretation of
within-group variability observed in ELISA validation.

(1) About AUC source and performance metrics

**In the original manuscript, the AUC was derived from a predictive model built using 10 combined**
**biomarkers.** To clarify the model construction, a schematic of the machine learning workflow was
included in the revised version (**Figure RL2A**). We first randomly divided the cohort into training and
test sets in a 4:1 ratio, followed by feature selection performed on the training set. Ultimately, 10
diagnostic biomarkers were selected, including six upregulated proteins (DHX9, CUTA, LONP1,
BCAR1, SPARC, and ARHGEF10) and four downregulated proteins (ANXA2, MME, GNAI3, and
MANBA). The predictive model based on these 10 combined biomarkers exhibited robust classification

performance on the test set, achieving an AUC of 0.96. **In the revised version, we also reported**
**additional key performance metrics: an accuracy of 0.88, precision of 0.88, recall of 0.91, and an**
**F1 score of 0.88 (Figure RL2B).** These findings suggested the potential clinical application of the model
in the diagnosis of CP.

Thank you again for your professional and constructive comments, which have greatly helped improve
the quality of this manuscript. The related information was updated in **Figure 1D, 1F,** and **“Result”**
**section (lines 177-178 on pages 6)** in the revised manuscript.

**Figure RL2. (A)** The schematic outlines the machine learning framework. **(B)** Receiver Operating
Characteristic (ROC) curves evaluating the performance of the 10 combined biomarkers, accompanied
by confusion matrices illustrating the performance of the machine learning classifier.

(2) About the sample characteristics

We fully agree with the reviewer’s suggestion that providing detailed cohort descriptions enhances the
reproducibility and transparency of the study. **In response to the reviewer’s comment, we have**
**included detailed sample characteristics in the revised manuscript, including age and sex**
**distributions in both the CP and control groups.** Specifically, the median age was 23 months in the
CP group and 24 months in the control group. To further evaluate differences in age distribution between
the two groups, the Wilcoxon rank-sum test was performed, showing no significant difference ($P=0.16$)
**(Figure RL2C).** The sex distribution did not significantly differ between groups (CP: 69.2% male; HC:
74% male; $P = 0.233$, Fisher’s exact test) **(Figure RL2D).**

**Additionally, we also supplemented detailed clinical characteristics of the CP patients, including**
**clinical type, GMFCS classification, MRI subtypes, and known risk factors such as preterm birth**
**and low birth weight in the revised version (Figure RL2E).** Spastic CP was the most common subtype

(80.8%), and a higher proportion of cases were male (69.2%). These distribution patterns of CP types
 were consistent with previously reported epidemiological trends in CP (*JAMA Pediatr*, 2017, PMID:
 28715518; *Lancet Neurol*, 2023, PMID: 36657477). These sample characteristics provided a
 comprehensive clinical profile of the study cohort.

Thank you again for your professional and constructive comments, which have greatly helped improve
 the quality of this manuscript. The related information was updated in **Figure S1A-B** and “**Results**”
 section (**lines 133-141 on pages 5**) in the revised manuscript.

**Figure RL2.** (C) Age and sex distribution in the study cohort. Fisher’s exact test was used to compare
 sex distribution between the CP and HC groups, and the Wilcoxon rank-sum test was used to compare
 age distribution. (D) Density plot showing the age distribution in cohort. (E) Clinical characteristics of
 CP patients. The MRI Classification System (MRICS) was used to categorize brain MRI findings into
 five types: Maldevelopments, Predominant White Matter Injury, Predominant Gray Matter Injury,
 Miscellaneous, and Normal (*Dev Med Child Neurol*, 2017, PMID: 27325153).

(3) About the within-group variability

We thank the reviewer for the valuable comment. CP is a group of non-progressive neurodevelopmental
 disorders postulated to be due to early brain injury. Its complex etiology and diverse clinical
 manifestations pose substantial challenges to early diagnosis and the evaluation of biomarker stability.

To improve diagnostic accuracy, our study identified a panel of candidate biomarkers with potential
clinical utility based on plasma proteomic data, and validated their differential expression in an
independent cohort using ELISA. **As noted by the reviewer, some biomarkers exhibited relatively**
**high within-group variability in the ELISA validation (original manuscript: Figure 1H).** To answer
the reviewer's questions clearly, we divided the response into three parts.

**1) Identification of robust biomarkers for CP diagnosis.**

In this study, we collected serum samples from 346 children with CP and 190 healthy controls (HC) for
proteomic analysis. Statistical tests combined with machine learning algorithms were applied in the
training set to select discriminative proteins. As a result, 10 candidate biomarkers were identified, with
6 proteins significantly upregulated in the CP group (DHX9, CUTA, LONP1, BCAR1, SPARC, and
ARHGEF10) and 4 proteins significantly downregulated (ANXA2, MME, GNAI3, and MANBA). We
then constructed a multi-marker model to evaluate diagnostic performance in the test set. The model
showed high stability and discriminative ability in the testing set, achieving an AUC of 0.96, an accuracy
of 0.88, a precision of 0.88, a recall of 0.91, and an F1 score of 0.88.

To validate the cross-platform reproducibility of the candidate biomarkers, their expression levels were
further examined using ELISA in an independent validation cohort (CP=38, HC=32). The results showed
that all 10 biomarkers remained statistically significant on the ELISA platform ($P < 0.05$), indicating
potential for clinical utility (**Figure RL2F**). Notably, some biomarkers exhibited considerable within-
group variability in the ELISA validation. Individual protein expression levels might show variability
due to biological factors, a challenge commonly observed in studies of neurodevelopmental disorders
(Science, 2015, PMID:26472761). This observation also might be attributable to technical aspects of the
immunoassay platform, including its narrower dynamic range, which might affect the accuracy and
reproducibility of protein quantification (*Nat Biotechnol*, 2006, PMID: 16900146; *EMBO Mol Med*, 2019,
PMID: 31566909).

In summary, although a panel of candidate biomarkers with potential diagnostic value was identified
based on proteomic profiling, the within-group variability observed in the ELISA suggests that single-
marker strategies might face limitations in clinical applicability due to insufficient stability.

**Figure RL2. (F)** Boxplot showing the ELISA result of candidate biomarkers.

**2) Challenges in clinical utility of single biomarkers.**

Although individual biomarkers might show strong disease associations at the mechanistic level, their
 clinical utility is often limited by variability and insufficient robustness (*Sci Transl Med*, 2010, PMID:
 20739680). **First, a single molecule is often insufficient to capture the multidimensional**
 **pathological mechanisms of complex diseases.** For example, in CP and other neurodevelopmental
 disorders, studies have shown that disease onset and progression involve dysregulation across multiple
 pathways, including metabolic disturbances, impaired neural plasticity, and immune-inflammatory
 responses (*Nat Rev Dis Primers*, 2016, PMID: 27188686). Against this background, changes in the
 expression of a single protein may reflect only localized aspects of the disease, which may fail to capture
 the overall pathophysiology and thus introduce diagnostic bias or instability.

**Second, due to limitations in detection performance and inter-individual variability, a single**
 **marker often fails to achieve both high sensitivity and high specificity** (*Lancet Digit Health*, 2024,
 PMID: 39332854). In clinical settings, substantial individual differences—such as age and basal
 metabolic state—might all affect protein expression levels. Moreover, immunoassays such as ELISA
 have inherent limitations in parameters such as antibody specificity and detection range, which can
 compromise the accuracy and consistency of protein quantification, ultimately affecting the reliability of
 diagnostic results.

Taken together, these findings emphasized the limitations of single-marker approaches and supported the
 need for multi-marker strategies to improve diagnostic robustness considering biological heterogeneity.
 Previous studies have demonstrated that integrative modeling of multiple biomarkers can improve
 robustness, reduce susceptibility to noise, and enhance cross-platform applicability (*Nat Biotechnol*,
 2006, PMID: 16900146). **In this study, we thus employed a multi-marker panel to improve**
 **diagnostic performance under high biological variability, laying the foundation for future clinical**
 **validation.**

**3) Clinical implications of multi-biomarker panel optimization for CP diagnosis.**

**To better answer the reviewer’s question, we validated the diagnostic performance of the multi-**
**marker model based on ELISA results of 10 biomarkers.** The samples were first randomly divided
into a training set and a testing set at a 6:4 ratio. An XGBoost-based multi-marker prediction model was
then constructed using 10-fold cross-validation on the training set. The result showed that the model
maintained excellent classification performance in the testing set, with an AUC of 0.98, an accuracy of
0.93, a precision of 0.94, a recall of 0.94, and an F1 score of 0.94 (**Figure RL2G**), further supporting the
ability of the combined model to mitigate within-group variability and enhance diagnostic stability.

**To improve the clinical applicability of the biomarkers, we used logistic regression with the Youden**
**index to define an optimal diagnostic threshold.** This strategy is consistent with widely adopted
approaches in the literature and offers good generalizability and reproducibility (*Adv Sci*, 2021, PMID:
34026427; *Nat Med*, 2025, PMID: 40164724). Specifically, a logistic regression model was constructed
using the ELISA-derived expression levels of the 10 candidate biomarkers to calculate a risk score for
each participant. The optimal cutoff value was then determined using the Youden index, aiming to
achieve the best balance between sensitivity and specificity. The composite risk score was calculated
using the following formula:

$$\text{Risk Score} = w_0 + w_1 \cdot \text{Protein}_1 + w_2 \cdot \text{Protein}_2 + \dots + w_n \cdot \text{Protein}_n$$

In this formula, w_0 represents the intercept, while w_1, w_2, \dots, w_n are the weighted coefficients
corresponding to each biomarker. $\text{Protein}_1, \text{Protein}_2, \dots, \text{Protein}_n$ represent the log-transformed ELISA
expression levels of the 10 biomarkers.

**The results showed that the risk scores generated from the 10-biomarker model significantly**
**distinguished CP patients from healthy controls ($P < 0.05$) (Figure RL2H).** Using Youden index
analysis, the optimal diagnostic threshold for the combined model was determined to be 0.59. Individuals
with scores above this threshold were classified as CP, and those below as controls. At this cutoff, the
model demonstrated excellent diagnostic performance, with a Youden index of 0.95, sensitivity of 0.95,
and specificity of 1, suggesting potential for clinical diagnosis of CP.

**To maintain diagnostic performance while improving cost-efficiency and operational simplicity, we**
**further optimized the model by reducing the number of biomarkers.** Specifically, we systematically
evaluated all possible three-marker combinations (a total of 120 models) and calculated risk scores for
each. Among all 120 three-protein combinations, the panel of SPARC, DHX9, and MANBA achieved
the highest Youden index (0.88), with a sensitivity of 0.95, specificity of 0.94, and an optimal cutoff
value of 0.49 (**Figure RL2I and Table RL1-1**). This three-biomarker panel offered practical advantages
in implementation and broader applicability.

In conclusion, the multi-marker model developed in this study, combined with a risk score strategy,
effectively reduced the influence of single-marker variability and significantly enhanced diagnostic
stability and clinical applicability. Furthermore, given the current limitations of commercial antibodies

in terms of specificity, sensitivity, and batch consistency—especially in heterogeneous diseases such as
 CP—future research should focus on developing high-affinity antibodies for key targets and conducting
 large-scale, multicenter validation to support early clinical implementation.

Thank you again for your professional and constructive comments, which have greatly helped improve
 the quality of this manuscript. The related information was updated in **Figure 1G-H and S1L**, “**Results**”
 **section (lines 203-216 on pages 7)** and “**Methods**” **section (lines 680-690 on pages 23)** in the revised
 manuscript.

**Figure RL2. (G)** Receiver Operating Characteristic (ROC) curve evaluating the performance of
 biomarkers based on ELISA results in an independent cohort, accompanied by a confusion matrix
 illustrating the performance of the machine learning classifier. **(H)** Boxplot displaying the risk score
 distribution based on the 10 biomarkers derived from ELISA results. **(I)** Boxplot showing the risk score
 based on SPARC-DHX9-MANBA panel.

Table RL1-1. The top 10 three-marker combinations ranked by the Youden index.

Three-marker combinations	Best_Cutoff	Youden_Index	Sensitivity	Specificity
SPARC-DHX9-MANBA	0.48626562	0.884868421	0.94736842	0.9375
DHX9-GNAI3-MANBA	0.418961628	0.875	1	0.875
SPARC-GNAI3-MANBA	0.460223623	0.863486842	0.89473684	0.96875
CUTA-DHX9-MANBA	0.576648163	0.858552632	0.92105263	0.9375
DHX9-MANBA-MME	0.463008861	0.853618421	0.94736842	0.90625
ARHCEF10-GNAI3-MANBA	0.390671999	0.848684211	0.97368421	0.875
GNAI3-MANBA-MME	0.470427956	0.848684211	0.97368421	0.875
SPARC-ARHCEF10-MANBA	0.558889696	0.842105263	0.84210526	1
SPARC-MANBA-ANXA2	0.551741193	0.842105263	0.84210526	1
BCAR1-DHX9-MANBA	0.538775064	0.837171053	0.86842105	0.96875

**R1Q3. Sample Heterogeneity**

**Although the discussion mentions overcoming sample heterogeneity, specific details on data**
**processing steps to mitigate this variability would improve clarity and facilitate study replication.**

**Response to R1Q3:**

We sincerely thank the reviewer for this valuable comment and apologize for the inaccurate use of the
term “overcome heterogeneity” in the original manuscript. CP exhibits substantial individual variability
due to its complex etiology and diverse clinical phenotypes. This biological and phenotypic heterogeneity
is common in clinical research and often unavoidable, posing significant challenges to related studies. **In**
**the original manuscript, we implemented standardized procedures across key steps—including**
**sample collection, processing, and data analysis—to reduce technical and operational bias and**
**ensure robust biomarker identification.** To answer the reviewer’s questions clearly, we divided the
response into three parts.

**(1) Sample collection**

**1) Standardized Clinical Diagnosis**

The diagnostic criteria for CP were consistent with our previous study (*Nat Med*, 2024, PMID: 38693247).
**All children diagnosed with CP met Surveillance of Cerebral Palsy in Europe (SCPE) criteria (*Dev***
***Med Child Neurol Suppl*, 2000, PMID: 11132255).** Inclusion and exclusion criteria were as follows:

- • The inclusion criteria: 1) Diagnosed with CP following comprehensive clinical evaluations
conducted by pediatric neurologists, including pregnancy and birth history, family history, and
detailed physical and neurological examinations; 2) No known chromosomal abnormalities or
syndromic diagnoses. 3) All patients were under 18 years of age. For those who received an initial
diagnosis before the age of 2, longitudinal follow-up was performed until they reached at least 2
289 years of age to confirm the stability and accuracy of the diagnosis (*Nat Med*, 2024, PMID:
38693247).
- • The exclusion criteria: 1) Unstable vital signs or serious systemic illness; 2) CP secondary to
acquired causes (e.g., encephalitis, uncontrolled epilepsy); 3) CP subtypes predominantly
characterized by hypotonia or rigidity.

Healthy control samples were obtained during routine pediatric medical examinations at the Third
Affiliated Hospital of Zhengzhou University, a major hospital responsible for standardized health
screening of infants and children in Henan Province, China. Parents or legal guardians of all children
included in this study were counseled and provided written informed consent for participation. This
consent encompassed the use of biological samples for other approved biomedical research. The
diagnostic criteria were consistent with our previous studies (*Mol Psychiatry*, 2024, PMID: 38762692;
*Brain Behav Immun*, 2023, PMID: 37011865). Inclusion and exclusion criteria were as follows:

- • **The inclusion criteria:** 1) **Typically developing children with no history of neurological,**
**genetic, or developmental disorders.** 2) No recent perinatal complications or current medications;
- • **The exclusion criteria:** 1) Recent infections, fever, or vaccination within two weeks; 2) First-
degree family history of neurodevelopmental or inherited genetic disorders;

The related information was updated in “Methods” section (lines 494-513 on pages 17) in the revised
manuscript.

2) Serum sample collection and preservation

**Standardized serum sample collection and preservation are essential for mitigating the effects of**
**pre-analytical variability on sample quality.** In this study, all serum samples were collected by certified
clinicians at affiliated hospitals of Zhengzhou University after obtaining informed consent, and
procedures were strictly **performed in accordance with the Guidelines for Venous Blood Specimen**
**Collection (WS/T 661-2020) to standardize collection and improve inter-sample consistency.**
Specifically, blood samples were drawn using dry vacuum tubes without anticoagulants, allowed to clot
at room temperature, and then centrifuged at $1,000 \times g$ for 10 minutes at 4°C to separate the serum. All
samples were processed within 2 hours of collection, and the serum stored at -80°C until further analysis.

This standardized processing and storage protocol has been applied in multiple serum proteomic studies
and has been shown to effectively reduce the impact of long-term storage on sample stability and data
reliability (*EMBO Mol Med*, 2022, PMID: 34978375; *Cell*, 2020, PMID: 32492406).

The related information was updated in “Methods” section (lines 515-521 on pages 17-18) in the revised
manuscript.

3) Cohort design and consistency with epidemiological trends

**To reduce potential confounding, we selected CP and healthy participants with similar age and sex**
**distributions, which were confirmed to be statistically non-significant.** Specifically, the median age
was 23 months in the CP group and 24 months in the control group. A Wilcoxon rank-sum test indicated
no significant difference in age distribution ($P=0.16$) (Figure RL3A). The sex distribution did not
significantly differ between groups (CP: 69.2% male; HC: 74% male; $P = 0.233$, Fisher’s exact test)
(Figure RL3B).

**Moreover, the CP cohort featured diverse clinical profiles, including motor subtypes, GMFCS**
**levels, MRI classifications, and known risk factors (Figure RL3C).** The spastic CP was the most
common subtype (80%), and a higher proportion of cases were male (70%). These distribution patterns
align well with previously reported epidemiological trends in CP (*JAMA Pediatr*, 2017, PMID: 28715518;
*Lancet Neurol*, 2023, PMID: 36657477). The cohort also included common perinatal risk factors such as
preterm birth and low birth weight. **These data collectively contributed to a systematic**
**characterization of clinical and enhance the representativeness and reproducibility of our findings.**

The related information was updated in Figure S1A-B and “Results” section (lines 133-141 on pages
5) in the revised manuscript.

A

Characteristics	CP (n = 346)	HC (n = 190)	P
Sex			
Female	106 (30.8%)	49 (26%)	0.233
Male	238 (69.2%)	141 (74%)	
Unknown	2		
Age(month)	23	24	0.16

B

C

Characteristics	CP	Risk factors	CP
Type		Adverse pregnancy history	21 (6.1%)
Ataxic	9 (2.7%)	Threatened abortion	45 (13%)
Dyskinetic	17 (4.9%)	Multiples pregnancy	35 (10.1%)
Mixed	40 (11.6%)	Placental abnormality	15 (4.3%)
Spastic	278 (80.8%)	Fetal distress	6 (1.7%)
Unknown	2	Pregnancy-related complications	58 (16.8%)
GMFCS		Umbilical cord around neck	58 (16.8%)
level I	90 (26.3%)	Abnormal amniotic fluid	45 (13%)
level II	80 (23.3%)	Perinatal asphyxia	93 (26.9%)
level III	60 (17.4%)	Low birth weight	101 (29.2%)
level IV	47 (13.7%)	Preterm birth	118 (34.1%)
level V	63 (18.3%)	PVL	130 (37.6%)
Unknown	2	Pathological jaundice	102 (29.5%)
MRICS		Intracranial hemorrhage	17 (4.9%)
Maldevelopments	18 (6.1%)	Neonatal hypoglycemia	27 (7.8%)
Predominant White Matter Injury	183 (61.8%)		
Predominant Gray Matter Injury	43 (14.5%)		
Miscellaneous	29 (9.8%)		
Normal	23 (7.8%)		
Unknown	50		

**Figure RL3. (A)** Age and sex distribution in the study cohort. Fisher's exact test was used to compare
 sex distribution between the CP and HC groups, and the Wilcoxon rank-sum test was used to compare
 age distribution. **(B)** Density plot showing the age distribution in cohort. **(C)** Clinical characteristics of
 CP patients. The MRI Classification System (MRICS) was used to categorize brain MRI findings into
 five types: Maldevelopments, Predominant White Matter Injury, Predominant Gray Matter Injury,
 Miscellaneous, and Normal (*Dev Med Child Neurol*, 2017, PMID: 27325153).

(2) Sample processing and Quality Control

1) Protein concentration normalization

**To minimize systematic errors caused by variations in protein concentration, all samples were**
 **adjusted to a uniform protein concentration before undergoing mass spectrometry.** Specifically,
 total protein levels were measured using a NanoDrop spectrophotometer (Thermo Fisher Scientific) at
 280 nm and adjusted to same concentration. Finally, 200 ng of tryptic peptides from each sample was
 analyzed by mass spectrometry to ensure cross-sample comparability. The related information was in
 "Methods" section (lines 556-558 on pages 19) in the revised manuscript.

2) QC quality control

We employed a mass spectrometry-based, high-throughput data-independent acquisition (DIA)

quantitative proteomics approach to identify serum proteins (*Nat Commun*, 2024, PMID: 38719824; *Nat*
 *Commun*, 2024, PMID: 38302471). **To ensure the stability of the reproducibility of the entire**
 **experimental process and the mass spectrometry platform performance, two types of quality**
 **control (QC) samples were established** in this study, consistent with previous studies (*Cell Syst*, 2016,
 PMID: 27135364):

- • **HEK293T QC:** Lysates from the human embryonic kidney cell line HEK293T, provided by the
 National Infrastructure Cell Line Resource, were used to continuously monitor the operational
 stability of the LC-MS/MS (*Cell Res*, 2022 PMID: 36307579).
- • **Pooled QC:** The quality control samples were prepared by mixing all serum samples in equal
 proportions and were used to systematically evaluate the entire experimental workflow—from
 sample preparation to mass spectrometry analysis—ensuring the stability and reliability of the data
 (*Cell*, 2020, PMID: 32492406).

 The preparation of pooled QC samples followed the same protocol as the cohort serum samples. Mass
 spectrometry assays for both the pooled QC and HEK293T samples were aligned with those of the cohort
 samples to ensure consistency. The HEK293T QC were analyzed twice daily, while pooled QC samples
 were measured once every 48 sample runs. This resulted in a total of 20 HEK293T QC runs and 11
 pooled QC runs. The results showed that the average Spearman correlation coefficient among pooled QC
 samples was 0.97, indicating high technical reproducibility of the experimental workflow (**Figure RL3**
 **D**). For the HEK293T QC samples, the average Spearman correlation coefficient reached 0.98, reflecting
 excellent operational stability and analytical performance of the LC-MS/MS platform.

 The related information was in **Figure S1C**, “**Results**” section (lines 143-151 on pages 5), and
 “**Methods**” section (lines 591-598 on pages 20) in the revised manuscript.

**Figure RL3. (D)** Spearman’s correlation analysis of HEK293T cell QC and serum pooled QC.

**(3) Data processing and diagnostic biomarker selection**

To improve the robustness of downstream analysis, we implemented a standardized preprocessing
workflow for proteomic data, which included normalization and imputation steps (*Cell Res*, 2022, PMID:
36307579; *Cell Metab*, 2025, PMID: 39488213). **First**, we used the fraction of total (FOT) method to
normalize protein-level quantification data by dividing the intensity of each protein by the total intensity
of all proteins within the same sample. protein-level quantification data. This approach helps account for
variations in total protein content across samples and offers a standardized measure of relative protein
abundance. It has been adopted in multiple published studies for proteomic normalization (*Journal of*
*Hematology & Oncology*, 2022, PMID: 35659036; *Cell Res*, 2022, PMID: 36307579).

**In addition, to more reliably identify biomarkers with stable diagnostic performance, we**
**established a standardized feature selection process and validated their expression stability and**
**diagnostic consistency using ELISA in an independent cohort.** Firstly, to ensure data quality, proteins
with less than 50% missing values were retained (*Cell Metab*, 2025, PMID: 39488213;). Subsequently,
**protein expression data were log₂-transformed** across all samples prior to machine learning model
construction. Secondly, the samples were randomly divided into training and testing sets, and feature
selection was strictly performed on the training set to ensure the validity and generalizability of model
evaluation (*Nature*, 2022, PMID: 34875674). Specifically, we identified differentially expressed proteins
through linear regression adjusted for age and sex, then evaluated their predictive importance using
XGBoost-based feature ranking. Finally, a multi-marker diagnostic model was constructed using the top
10 most important feature proteins, achieving an AUC of 0.96 with accuracy of 0.88, precision of 0.88,
recall of 0.91, and an F1-score of 0.88.

**To validate the cross-platform reproducibility of the candidate biomarkers, their expression levels**
**were further examined using ELISA in an independent validation cohort** (CP=38, HC=32). The
results showed that all 10 biomarkers remained statistically significant on the ELISA platform ($P < 0.05$),
indicating their potential for clinical utility. **To better answer reviewer question, we further evaluated**
**the diagnostic performance of the 10-biomarker combination model using the ELISA dataset in**
**the revised version.** The samples were randomly divided into a training set and a testing set at a 6:4
ratio. An XGBoost-based multi-marker prediction model was then constructed using 10-fold cross-
validation on the training set. The result showed that the model maintained excellent classification
performance (AUC = 0.98; accuracy = 0.93; precision = 0.94; recall = 0.94; F1 score = 0.94) in testing
set, suggested robustness and cross-platform diagnostic consistency (**Figure RL3E**).

In summary, while biological heterogeneity is an inherent feature of CP due to its complex etiology and
diverse clinical manifestations, we implemented a series of standardized and systematic strategies to
reduce non-biological variability throughout the study. The consistent diagnostic performance across
platforms further supports the robustness and translational potential of the identified biomarker panel. In
future studies, we plan to expand the sample size to further validate the model's generalizability under
heterogeneous conditions. Additionally, the development of standardized detection kits based on these
biomarkers will be a critical step toward their clinical implementation in early screening.

The related information of data preprocessing was updated in the “**Methods**” section (**lines 586-589 on**

pages 20; lines 654-656 on pages 22) and the AUC of ELISA validation was updated in **Figure 1H** and
“Results” section (lines 203-205 on pages 7) in the revised manuscript.

**Figure RL3. (E)** Receiver Operating Characteristic (ROC) curve evaluating the performance of
biomarkers based on ELISA results in an independent cohort, accompanied by a confusion matrix
illustrating the performance of the machine learning classifier.

**R1Q4. Sample Classification**

**The criteria for classifying samples into genetic CP, non-genetic CP, and healthy controls require**
**clarification. For instance, are patients classified as genetic CP if they harbor at least one**
**pathogenic/likely pathogenic variant?**

**Response to R1Q4:**

We sincerely apologize for the unclear definitions of the sample classification (genetic CP, non-genetic
CP, and healthy controls) in the original manuscript. **In the revised manuscript**, according to the
reviewer’s insightful comments, we have provided a clearer and more detailed description of these
classifications in the **Methods section**. The detailed responses are as follows.

**(1) As for the Definition of genetic CP and non-genetic CP**

All patients underwent comprehensive clinical evaluations conducted by pediatric neurologists, and their
diagnoses conformed to the criteria defined by the Surveillance of Cerebral Palsy in Europe (SCPE) (*Dev*
*Med Child Neurol Suppl*, 2000, PMID: 11132255).

In the original manuscript, the variant results for 321 patients among 346 CP were obtained from our
previously study (*Nat Med*, 2024, PMID: 38693247). **The classification of patients into genetic and**
**non-genetic CP groups was based on the pathogenic and likely pathogenic (P/LP) variants**
**identified in that study.** In response to the reviewer’s comment, we have included a schematic overview
of the variant filtering strategy, as described in the original publication (**Figure RL4**). Briefly, variants
were annotated using ANNOVAR (v2020-06-08). Variants were retained if they were located in genes

listed in the OMIM and HGMD databases and associated with CP phenotypes. These variants were
 subsequently categorized as pathogenic (P), likely pathogenic (LP), of uncertain significance (VUS),
 benign (B), or likely benign (LB) according to ACMG/AMP guidelines. Because pathogenic and likely
 pathogenic (P/LP) variants are considered clinically significant and linked to disease susceptibility
 (*Genet Med*, 2015, PMID: 25741868), all such variants identified in the study were confirmed by Sanger
 sequencing. **Finally, individuals were classified as genetic CP if they carried at least one P/LP**
 **variant in a gene with a known inheritance pattern and a clinical phenotype consistent with the**
 **gene's associated disorder. The classification of non-genetic CP in this study refers to cases in which**
 **no pathogenic or likely pathogenic variants were identified.**

**Figure RL4.** Schematic diagram showing the selection process of genetic variants (*Nat Med*, 2024,
 PMID: 38693247).

**(2) As for the Definition of Healthy Control.**

Healthy control (HC) samples were obtained during routine pediatric medical examinations at the Third
 Affiliated Hospital of Zhengzhou University, a major hospital responsible for standardized health
 screening of infants and children in Henan Province, China. Parents or legal guardians of all children
 included in this study were counseled and provided written informed consent for participation. This
 consent encompassed the use of biological samples for other approved biomedical research. The
 diagnostic criteria were consistent with our previous studies (*Mol Psychiatry*, 2024, PMID: 38762692;
 *Brain Behav Immun*, 2023, PMID: 37011865). Inclusion and exclusion criteria were as follows:

- • **The inclusion criteria:** 1) Typically developing children with no history of neurological,
 genetic, or developmental disorders. 2) No recent perinatal complications or current medications;
- • **The exclusion criteria:** 1) Recent infections, fever, or vaccination within two weeks; 2) First-
 degree family history of neurodevelopmental or inherited genetic disorders;

The related information was updated in “**Methods**” section (lines 494-513 on pages 17; lines 524-536
on pages 18) in the revised manuscript.

**R1Q5. Clarification on Diagnostic Tools (Line 83)**

The statement, “While MRI enhances the diagnostic capability for CP, approximately 10% of
patients—particularly those with milder forms—exhibit normal MRI results,” may be somewhat
misleading. In clinical practice, CP diagnosis involves not only MRI but also assessments of brain
development and reflex examination. A rephrasing could better reflect the multifaceted diagnostic
approach.

**Response to R1Q5:**

We sincerely thank the reviewer for highlighting the importance of a multi-modal diagnostic approach
for CP. As noted in our manuscript, while MRI is critical for detecting structural brain abnormalities
(Lines 79-82), we fully agree that clinical diagnosis relies on comprehensive clinical evaluations (e.g.,
motor function assessment, neurological examinations). In our cohort, all CP diagnoses were established
by pediatric neurologists based on the SCPE criteria (Dev Med Child Neurol Suppl, 2000, PMID:
11132255) and comprehensive clinical assessments, including neuroimaging when available.

To clarify this, we have revised the sentence to better reflect the comprehensive approach to CP diagnosis.

**The revised version:** “In clinical practice, CP diagnosis typically involves a combination of neurological
assessments, developmental evaluations, and neuroimaging techniques such as magnetic resonance
imaging (MRI). Although MRI enhances diagnostic sensitivity by revealing structural abnormalities,
approximately 10% of CP patients show normal MRI findings, underscoring the limitations of relying
on imaging alone.”

Thank you again for your professional and constructive comments, which have greatly helped improve
the quality of this manuscript. The related information was updated in “**Introduce**” section (lines 80-85
on pages 3) in the revised manuscript.

**R1Q6. Age Range in CP Diagnosis (Line 136)**

The authors report a wide age range at CP diagnosis, with a mean of 29.6 months (ranging from 2
to 180 months). Did the authors analyze and describe the influence of age on CP diagnosis or
progression? This would provide valuable context for interpreting the findings.

**Response to R1Q6:**

Thank you for your valuable feedback. We sincerely apologize for the lack of clarity in the description
of the influence of the age on the diagnosis of CP in the original manuscript. As increasing evidence
indicates that age is closely associated with disease occurrence and overall health status, adequately
accounting for age is essential in disease-related research. **Therefore, we had adjusted for the potential
impact of age on differentially expressed proteins and diagnostic biomarkers in the original**

**manuscript.** To answer the reviewer’s questions clearly, we divided the response into two parts: (1) As
for assessing the age distribution in our cohort; (2) As for assessing the age effects on diagnostic
biomarkers;

**(1) As for assessing the age distribution in our cohort.**

CP is a non-progressive neurodevelopmental disorder that originates from brain injury during the fetal
or infant period. Its complex etiology and highly heterogeneous clinical phenotype not only pose
diagnostic challenges but also result in a varying age range at diagnosis. Our study cohort showed that
the age at diagnosis ranged from 2 to 180 months. **The diagnosis of CP in children under two years of
age is particularly challenging** (*JAMA Pediatr*, 2017, PMID: 28715518). This diagnostic challenge
might be attributed to several factors, including poor patient cooperation during neurological assessments
and significant interindividual variability in early motor patterns, both of which increase diagnostic
uncertainty. Additionally, due to nervous system immaturity, motor development differences between
children with CP and typically developing peers may be subtle, further complicating early identification.
Therefore, international guidelines (NICE guideline NG62) recommend close monitoring of suspected
cases until 2 years of age to confirm the diagnosis. Given the diagnostic challenges of CP—especially in
early childhood and across diverse age groups—there is an urgent need for objective, broad-spectrum
biomarkers, such as blood-based proteomic markers, to enable accurate and timely diagnosis.

To address this gap, we collected serum samples from 346 individuals with CP and 190 age-matched
healthy controls (HC). All children diagnosed with CP met SCPE criteria (*Dev Med Child Neurol Suppl*,
2000, PMID: 11132255). To ensure diagnostic accuracy, all cases of **CP diagnosed before 2 years of
age underwent longitudinal follow-up until 2 years of age to confirm the diagnosis** (*Nat Med*, 2024,
PMID: 38693247). The final diagnostic classification and clinical information used in this study were
determined after the age of 2 years for these early-diagnosed cases.

To better answer the reviewer’s question, we further showed the age distribution in CP and HC groups.
**Density plots indicated that the median ages of the CP and HC groups were 23 and 24 months,
respectively. Statistical testing revealed no significant difference in age distribution between the
two groups (P = 0.16) (Figure RL5A).** These findings suggested that the overall age distribution was
matched between groups, reducing the likelihood of age-related bias in downstream analyses.
Importantly, the broad age range of our cohort also enabled assessment of biomarker applicability across
different stages of neurodevelopment.

**(2) As for assessing the age effects on diagnostic biomarkers in our cohort.**

**Given that robust biomarkers should be independent of age, we included age as a covariate in the
original analysis to control for its potential confounding effect.** Specifically, we applied a multiple
**linear regression model to account for age-related confounding** (R limma package 3.50.3 version)
(*Science*, 2024, PMID: 38781393; *Nat Genet*, 2025, PMID: 39972214). To further control for potential
confounding factors, sex was additionally incorporated into the model as covariates (Protein expression
~ Group + Age + Sex + intercept). **Notably, the diagnostic biomarkers reported in the original
manuscript were identified after adjusting for age covariate. To further assess the influence of age**

**on the selected diagnostic markers, we compared the regression coefficients for disease and age in**
**the multivariate model in the revised manuscript.** Notably, the correlation of all biomarkers with
disease was significantly stronger than that with age, suggesting that age had minimal influence on the
expression of these diagnostic markers (**Figure RL5B**).

**To evaluate the diagnostic robustness of our biomarker panel across developmental stages, we**
**stratified the cohort into two age groups: ≤ 24 months and > 24 months.** The 24-month threshold was
selected based on clinical relevance, as the diagnosis of CP typically becomes more stable after 2 years
of age (*JAMA Pediatr*, 2017, PMID: 28715518). This stratification allows us to assess whether biomarker
performance at early diagnosis remains consistent (≤ 24 months). The ≤ 24 -month group included 298
individuals (196 CP, 102 HC), and the > 24 -month group included 237 individuals (148 CP, 89 HC).
Within each subgroup, samples were randomly divided 4:1 into training and test sets. Using the same
panel of 10 biomarkers, we trained XGBoost models with 10-fold cross-validation. **The model achieved**
**an AUC of 0.96 (accuracy=92%, precision=93%, recall=92%, and F1-score=91%) in the ≤ 24**
**months group and 0.94 (accuracy=81%, precision=82%, recall=81%, and F1-score=81%) in**
**the > 24 months group (Figure RL5D-E).** These findings supported the robustness and general
applicability of our biomarker panel across different developmental stages, with consistent
diagnostic performance in both the ≤ 24 -month and > 24 -month groups, highlighting its potential utility
in early diagnosis and timely clinical intervention.

In summary, **our study systematically accounted for the potential confounding effects of age in both**
**cohort design and data analysis.** Furthermore, the diagnostic performance of our biomarker panel was
validated across different developmental stages, demonstrating its generalizability and clinical relevance.
**Notably, the combined biomarker panel also showed significant diagnostic performance in the CP**
**under two years of age, suggesting its potential role in early diagnosis.** Future studies with larger
sample sizes, including more infants with suspected early-stage cerebral palsy, are warranted to further
validate its diagnostic effectiveness.

The related information was updated in **FigureS1A, S1H, “Results” section (lines 139-141 and 176 on**
**pages 5-6), and “Methods” section (lines 659-662 on pages 22)** in the revised manuscript.

**Figure RL5.** (A) Density plot showing the distribution of age in cohort. (B) Bar plot illustrating the
 regression coefficients of Age and disease. (C-D) Receiver Operating Characteristic (ROC) curve
 evaluating the performance of biomarkers in ≤ 24 months and >24 months groups, accompanied by a
 confusion matrix illustrating the performance of the machine learning classifier.

**R1Q7. Minor Concerns:**

**The boxplot for EIF3L in Figure 1H is incorrect, as it lacks the sample from the HC group.**

**Response to R1Q7:**

We sincerely apologize for the omission of the EIF3L sample from the HC group in Figure 1H. We
 appreciate the reviewer's attention to this issue. **In the revised manuscript, we have corrected this**

**error and included the missing data for EIF3L in the boxplot.** To ensure the accuracy and consistency
of all figures, we have thoroughly reviewed the entire manuscript and verified all data visualizations to
prevent similar issues.

**Figure RL6** Boxplot showing the ELISA result of EIF3L.

**Reviewer #2 (Remarks to the Author):**

**The strengths of the study are:**

**Xu and colleagues conducted a proteomic analysis of serum samples from children with cerebral**
**palsy (CP) and healthy controls to identify potential diagnostic biomarkers for CP and to explore**
**the molecular mechanisms underlying the interplay of genetic and environmental risk factors. This**
**study recruited 346 children with CP and 190 healthy children, providing a robust dataset to**
**support the research. The substantial effort invested by the researchers in this study deserves**
**commendation, as it represents a notable strength of their work. The findings of this study are**
**expected to offer valuable insights for the early diagnosis and intervention of CP, with significant**
**clinical implications and practical application value.**

**The weaknesses of the study are:**

**Based on a comprehensive reading of the manuscript, the primary issue appears to be a lack of**
**refinement in the logical flow of the narrative. The main body of the text seems to be divided into**
**two parts: the first part focuses on the process of identifying and validating diagnostic biomarkers**
**(up to line 181), while the latter part emphasizes the exploration of the molecular mechanisms**
**linking genetic factors, environmental risk factors, and cerebral palsy. In my view, the connection**
**between these two sections is insufficiently developed in the writing.**

**Additionally, there are some questions regarding specific details in the manuscript:**

**Response to Reviewer #2:**

We sincerely appreciate the reviewer for the insightful comments and the positive feedback on the value
and significance of our findings. The reviewer's comments undoubtedly helped us improve the clarity of
our manuscript. In response to the reviewer's insightful comments, we have systematically revised the
manuscript: (1) improving the overall clarity and language quality, (2) conducting additional analyses to
assess the robustness of the diagnostic performance, (3) Adding detailed inclusion / exclusion criteria of
CP and healthy control, and (4) clarifying the classification criteria for genetic and non-genetic forms of
CP. All changes in the manuscript have been marked in red for clarity. The detailed point-by-point
responses were provided as follows.

**Introduction Section**

**R2Q1. The introduction of this paper highlights that MRI is currently an important auxiliary**
**method for diagnosing CP, but approximately 10% of mild CP patients exhibit normal MRI results.**
**Does the CP cohort in this study include such patients? If so, was it possible to validate these**
**biomarkers in this subgroup and achieve results consistent with the current findings? Such**
**validation would further demonstrate the superiority of the proposed diagnostic biomarkers.**

**Response to R2Q1:**

We appreciate the reviewer's insightful comments regarding the diagnostic performance of biomarkers
in patients with normal MRI findings. MRI is a valuable tool for the diagnosis and management of CP,
offering detailed information on brain injury across different regions (*JAMA*, 2006, PMID: 17018805).
However, previous studies have reported that approximately 10% of patients with cerebral palsy show
no detectable abnormalities on MRI (*Nat Rev Dis Primers*, 2016, PMID: 27188686; *JAMA Pediatr*, 2017,

PMID: 28715518). The presence of such cases increases the difficulty of early diagnosis and indicates
the need for additional diagnostic approaches beyond neuroimaging.

The MRI Classification System (MRICS) was used to categorize brain MRI findings into five types:
Maldevelopments, Predominant White Matter Injury, Predominant Gray Matter Injury, Miscellaneous,
and Normal (*Dev Med Child Neurol*, 2017, PMID: 27325153). **In our cohort, 273 patients had**
**abnormal MRI results, while 23 patients had normal MRI findings (Figure RL7A). To further**
**validate the stability of the biomarkers in patients with normal MRI findings, we conducted an**
**independent validation analysis of the MRI-normal subgroup in the revised version.** Specifically,
we constructed a balanced validation cohort by randomly selecting 23 individuals from the healthy
control (HC) group and pairing them with the 23 MRI-normal CP patients. Subsequently, we built a
classification model comprising 10 candidate biomarkers using the XGBoost algorithm with 10-fold
cross-validation. **The model demonstrated excellent diagnostic performance, achieving an AUC of**
**0.86, along with high accuracy (0.78), precision (0.78), recall (0.78), and F1 score (0.78) (Figure**
**RL7B).**

**These results indicated that the biomarker maintains stable and significant discriminatory ability**
**in MRI-normal patients, providing a valuable supplementary diagnostic tool for MRI-normal**
**population.** We fully agree with the reviewer's insight that this validation substantially enhances the
clinical translation potential of the biomarker. In the future studies, we plan to further validate and
strengthen the robustness of our findings by expanding the sample size, particularly by increasing the
inclusion of MRI-normal CP patients.

The related information was updated in **Figure S1K** and **“Result”** section (**lines 191-198 on pages 7**) in
the revised manuscript.

**Figure RL7.** (A) The distribution of MRI classifications within the study cohort. (B) Receiver Operating
Characteristic (ROC) curve evaluating the performance of the 10 combined biomarkers in MRI-normal
CP patients, accompanied by a confusion matrix illustrating the performance of the machine learning
classifier.

**Results Section**

**R2Q2. In the second part of the results, differentially expressed proteins (DEPs) between the CP**
**and healthy groups were analyzed, identifying 82 upregulated and 83 downregulated proteins.**
**However, in the machine learning analysis, 24 DEPs were selected as candidate proteins for the**
**training set. What were the criteria for this selection? (lines 171–173)**

**Response to R2Q2:**

We thank the reviewer for this valuable comment. We sincerely apologize for not making this point clear
in the previous version. **To prevent data leakage (i.e., unintentional use of information from the**
**testing set during model development) and enhance model generalizability, diagnostic protein**
**selection was conducted independently within the training set using a combination of statistical**
**testing and machine learning.** Therefore, the 24 differentially expressed proteins (DEPs) used for model
construction were not directly derived from the 165 DEPs identified in the global differential analysis.
To improve transparency and readability, we have provided a more detailed explanation below.

**(1) Global differential expression analysis (165 DEPs).**

To explore global proteomic alterations in CP patients, we conducted differential expression analysis
using all 536 samples (346 CP and 190 HC). This identified 165 DEPs (82 upregulated, 83 downregulated;
FDR < 0.05). **This analysis aimed to provide a comprehensive and unbiased overview of proteomic**
**differences between CP and HC.** This step was conducted **independently of machine learning**
**analysis** and served as an exploratory approach to identify biological pathways altered in CP.

**(2) Machine learning-based feature selection (24 DEPs).**

**To construct a robust and generalizable diagnostic model, we followed standard machine learning**
**procedures in the original manuscript** (*Patterns (N Y)*, 2023, PMID: 37720327; *Nat Methods*, 2024,
PMID: 39122953). **(Figure RL8A).** To enhance clarity, we had updated the workflow diagram in the
revised manuscript. Specifically, 536 samples were randomly split into a training set (n = 429) and a test
set (n = 107) at a 4:1 ratio. The training set was used for model building, while the test set was reserved
for validation. **To ensure methodological rigor and avoid data leakage, the selection of diagnostic**
**biomarkers was conducted exclusively within the training set** (*Nat Methods*, 2024, PMID: 39122953;
*Nat Commun*, 2024, PMID: 38395893).

**In the training set, we employed feature selection strategy consistent with previously study** (*Nature*,
2022, PMID: 34875674). Specifically, to reduce redundancy and improve model stability and
interpretability, we first removed highly correlated proteins, retaining only the most representative
features with strong associations to disease status. Differential expression analysis was then performed
on the remaining proteins, yielding 24 DEPs with FDR < 0.05. **Although this selection process was**
**conducted independently of the global differential analysis, it is noteworthy that 22 of the 24**
**selected proteins overlapped with the 165 DEPs identified from the full cohort, further supporting**
**their biological relevance and robustness.** In the final step, we applied an XGBoost-based feature
ranking algorithm (a tree-based machine learning algorithm commonly used for feature selection) to
assess the discriminative importance of these DEPs. The top 10 most informative proteins were selected

for model building. This process prioritized features that contributed most to classification performance.
The final diagnostic model based on these 10 biomarkers achieved strong performance on the test set,
with an AUC of 0.96, accuracy of 0.88, precision of 0.88, recall of 0.91, and F1 score of 0.88.

In summary, to ensure objective feature selection and prevent information leakage, the 24 DEPs were
independently selected from the training set using a standardized machine learning pipeline. This
approach ensured both statistical robustness, thereby supporting the development of a reliable diagnostic
model.

Thank you again for your valuable comments, which has helped us improve the clarity and completeness
of our data presentation. The related information was updated in “Results” section (lines 170-173 on
pages 6) and “Method” section (lines 654-666 on pages 22) in the revised manuscript.

A

**Figure RL8. (A)** The schematic outlines the machine learning framework.

**R2Q3. The study identified a panel of 10 candidate protein biomarkers through machine learning**
**and validated them in an independent cohort. However, Figure 1 only presents ELISA results for**
**9 proteins. What about the results for the remaining protein?**

**Response to R2Q3:**

We sincerely apologize for not presenting the ELISA results for all 10 diagnostic biomarkers. In the
revised version, we have updated the ELISA validation results to include six upregulated proteins (DHX9,
CUTA, LONP1, BCAR1, SPARC, and ARHGEF10) and four downregulated proteins (ANXA2, MME,
GNAI3, and MANBA) in CP (Figure RL9). The ELISA results for these proteins were consistent with
the trends observed in the proteomic data, suggesting the robustness of biomarker selection.

Thank you again for your valuable comments, which has helped us improve the clarity and completeness
of our data presentation. The related information was updated in Figure 1G in the revised manuscript.

ELISA result in an independent cohort

**Figure RL9 (A)** Boxplot showing the ELISA result of candidate biomarkers.

**R2Q4.** According to the results in Figure 1H, while the differences in protein expression are
 statistically significant, the magnitude of these differences between the CP and control groups
 appears small. What is your perspective on the clinical implications of such findings for diagnosis,
 treatment, or prognosis assessment?

**Response to R2Q4:**

We appreciate the reviewer's thoughtful comments. Identifying robust and clinically relevant biomarkers
 is critical for improving CP diagnosis. In the original manuscript, we identified potential biomarkers
 through proteomic analysis and subsequently validated their significance using ELISA in an independent
 cohort. **As noted by the reviewer, although these biomarkers showed statistically significant**
 **differences, several exhibited only modest expression changes in the ELISA validation.** To answer
 the reviewer's questions clearly, we divided the response into four parts: (1) Identification of robust
 biomarkers for CP diagnosis; (2) Challenges in clinical utility of single biomarkers; (3) Clinical
 implications of multi-biomarker panel optimization for CP diagnosis. (4) Clinical implications of
 diagnostic biomarkers for CP treatment and prognosis assessment.

**(1) Identification of robust biomarkers for CP diagnosis.**

The diagnosis of CP primarily relies on neuroimaging and neurodevelopmental assessments (*Nat Rev*
 *Dis Primers*, 2016, PMID: 27188686). The lack of reliable laboratory biomarkers further complicates
 early diagnosis, increasing the risk of both false positives and false negatives, which could delay access
 to early therapeutic strategies (*JAMA Pediatr*, 2017, PMID: 28715518; *JAMA Pediatr*, 2023, PMID:
 36648935).

In the original manuscript, the collected serum samples were initially analyzed by liquid
 chromatography-tandem mass spectrometry (LC-MS/MS). Through a two-step process of statistical

testing and feature selection, ten candidate biomarkers were then identified, including 6 upregulated (e.g.,
 DHX9, CUTA, LONP1, BCAR1) and 4 downregulated (e.g., ANXA2, MME, GNAI3, MANBA)
 proteins in CP patients. Based on these candidate biomarkers, we further constructed a multi-marker
 predictive model, showing high stability and discriminative ability in the testing set, achieving an AUC
 of 0.96, an accuracy of 0.88, a precision of 0.88, a recall of 0.91, and an F1 score of 0.88.

To validate the cross-platform reproducibility of the candidate biomarkers, their expression levels were
 further examined using ELISA in an independent validation cohort (CP=38, HC=32). The results showed
 that all 10 biomarkers remained statistically significant on the ELISA platform ($P < 0.05$) (**Figure**
 **RL10A**), indicating potential for clinical utility. Notably, the magnitude of expression changes in ELISA
 validation was relatively modest compared to the proteomic results. **This may be attributed to the**
 **narrower detection range of ELISA compared to mass spectrometry** (*Nat Biotechnol*, 2006, PMID:
 16900146; *Nat Rev Dis Primers*, 2016, PMID: 27188686; *EMBO Mol Med*, 2019, PMID: 31566909).
 These observations were consistent with previous studies in which ELISA detected smaller changes in
 protein expression compared to mass spectrometry, yet still provided meaningful diagnostic insights
 across various diseases (*JAMA Neurol*, 2017, PMID: 27992627; *Immunity*, 2020, PMID: 33128875; *Ann*
 *Rheum Dis*, 2019, PMID: 31005900). **Notably, many well-established clinical biomarkers exhibit**
 **only modest but clinically significant expression changes.** For example, although HbA1c levels $<5.7\%$
 are considered normal, a threshold of $\geq 6.5\%$ is clinically used to diagnose diabetes (*BMJ*, 2006, PMID:
 16974013). Similarly, a modest increase in serum creatinine (≥ 0.3 mg/dL) is a key diagnostic criterion
 for acute kidney injury (*Kidney Int*, 2010, PMID: 20706215) (*Kidney Int*, 2010, PMID: 20706215).

In summary, although a panel of candidate biomarkers with potential diagnostic value was identified
 based on proteomic profiling, the modest expression differences suggested that single-marker strategies
 might face limitations in clinical applicability due to insufficient stability.

**Figure RL10 (A)** Boxplot showing the ELISA result of candidate biomarkers.

**(2) Challenges in clinical utility of single biomarkers.**

Although individual biomarkers may show strong disease associations at the mechanistic level, their
clinical utility is often limited by variability and insufficient robustness (*Sci Transl Med*, 2010, PMID:
20739680). **First, a single molecule is often insufficient to capture the multidimensional**
**pathological mechanisms of complex diseases.** For example, in CP and other neurodevelopmental
disorders, studies have shown that disease onset and progression involve dysregulation across multiple
pathways, including metabolic disturbances, impaired neural plasticity, and immune-inflammatory
responses (*Nat Rev Dis Primers*, 2016, PMID: 27188686). Against this background, changes in the
expression of a single protein may reflect only localized aspects of the disease, which may fail to capture
the overall pathophysiology and thus introduce diagnostic bias or instability.

**Second, due to limitations in detection performance and inter-individual variability, a single**
**marker often fails to achieve both high sensitivity and high specificity** (*Lancet Digit Health*, 2024,
PMID: 39332854). In clinical settings, substantial individual differences—such as age and basal
metabolic state—can all affect protein expression levels. Moreover, immunoassays such as ELISA have
inherent limitations in parameters such as antibody specificity and dynamic range, which can
compromise the accuracy and consistency of protein quantification, ultimately affecting the reliability of
diagnostic results.

Taken together, these findings emphasized the limitations of single-marker approaches and supported the
need for multi-marker strategies to improve diagnostic robustness in the face of biological heterogeneity.
Previous studies have demonstrated that integrative modeling of multiple biomarkers could improve
robustness, reduce susceptibility to noise, and enhance cross-platform applicability (*Nat Biotechnol*,
2006, PMID: 16900146). **In this study, we therefore employed a multi-marker modeling approach**
**to enhance diagnostic performance under conditions of high biological variability, providing a**
**methodological foundation for future clinical validation and utility.**

**(3) Clinical implications of multi-biomarker panel optimization for CP diagnosis.**

To better answer the reviewer's question, **we validated the diagnostic performance of the multi-**
**marker model based on ELISA results of 10 biomarkers.** The samples were first randomly divided
into a training set and a testing set at a 6:4 ratio. An XGBoost-based multi-marker prediction model was
then constructed using 10-fold cross-validation on the training set. The result showed that the model
maintained excellent classification performance in testing set (AUC = 0.98; accuracy = 0.93; precision
= 0.94; recall = 0.94; F1 score = 0.94) (**Figure RL10B**), further supporting the ability of the combined
model to mitigate within-group variability and enhance diagnostic stability.

**To improve the clinical applicability of the biomarkers, we employed a logistic regression model in**
**combination with the Youden index to quantify a diagnostic threshold and enhance the**
**discrimination between CP and control groups.** This strategy is consistent with widely adopted
approaches in the literature and offers good generalizability and reproducibility (*Adv Sci*, 2021, PMID:
34026427; *Nat Med*, 2025, PMID: 40164724). Specifically, a logistic regression model was constructed
using the ELISA-derived expression levels of the 10 candidate biomarkers to calculate a risk score for

each participant. The optimal cutoff value was then determined using the Youden index, aiming to
achieve the best balance between sensitivity and specificity. The composite risk score was calculated
using the following formula:

$$\text{Risk Score} = w_0 + w_1 \cdot \text{Protein}_1 + w_2 \cdot \text{Protein}_2 + \dots + w_n \cdot \text{Protein}_n$$

In this formula, w_0 represents the intercept, while w_1, w_2, \dots, w_n are the weighted coefficients
corresponding to each biomarker. $\text{Protein}_1, \text{Protein}_2, \dots, \text{Protein}_n$ represent the log₂-transformed ELISA
expression levels of the 10 biomarkers.

**The results showed that the risk scores significantly distinguished CP patients from healthy**
**controls ($P < 0.05$) (Figure RL10C).** Using Youden index analysis, the optimal diagnostic threshold for
the combined model was determined to be 0.59. Individuals with scores above this threshold were
classified as CP, and those below as controls. At this cutoff, the model demonstrated excellent diagnostic
performance, with a Youden index of 0.95, sensitivity of 0.95, and specificity of 1, suggesting strong
potential for clinical diagnosis of CP.

**To maintain diagnostic performance while improving cost-efficiency and operational simplicity, we**
**further optimized the model by reducing the number of biomarkers.** Specifically, we systematically
evaluated all possible three-marker combinations (a total of 120 models) and calculated risk scores for
each. Among all 120 three-protein combinations, the panel of SPARC, DHX9, and MANBA achieved
the highest Youden index (0.88), with a sensitivity of 0.95, specificity of 0.94, and an optimal cutoff
value of 0.49 (Figure RL10D and Table RL2-1). This optimized panel model required only three
biomarkers, offering practical advantages in implementation and broader applicability.

In conclusion, the multi-marker model developed in this study, combined with a risk score strategy,
effectively reduced the influence of single-marker variability and significantly enhanced diagnostic
stability and clinical applicability. Furthermore, given the current limitations of commercial antibodies
in terms of specificity, sensitivity, and batch consistency—especially in heterogeneous diseases such as
CP—future research should focus on developing high-affinity antibodies for key targets and conducting
large-scale, multicenter validation to support early clinical implementation.

Thank you again for your professional and constructive comments, which have greatly helped improve
the quality of this manuscript. The related information was updated in Figure 1G-H and S1L, “Results”
section (lines 202-216 on pages 7) and “Methods” section (lines 680-690 on pages 23) in the revised
manuscript.

**Figure RL10.** (B) Receiver Operating Characteristic (ROC) curve evaluating the performance of
 biomarkers based on ELISA results in an independent cohort, accompanied by a confusion matrix
 illustrating the performance of the machine learning classifier. (C) Boxplot displaying the risk score
 distribution based on the 10 biomarkers derived from ELISA results. (D) Boxplot showing the risk score
 based on SPARC-DHX9-MANBA panel.

Table RL2-1. The top 10 three-marker combinations ranked by the Youden index.

Three-marker combinations	Best_Cutoff	Youden_Index	Sensitivity	Specificity
SPARC-DHX9-MANBA	0.48626562	0.884868421	0.94736842	0.9375
DHX9-GNAI3-MANBA	0.418961628	0.875	1	0.875
SPARC-GNAI3-MANBA	0.460223623	0.863486842	0.89473684	0.96875
CUTA-DHX9-MANBA	0.576648163	0.858552632	0.92105263	0.9375
DHX9-MANBA-MME	0.463008861	0.853618421	0.94736842	0.90625
ARHCEF10-GNAI3-MANBA	0.390671999	0.848684211	0.97368421	0.875
GNAI3-MANBA-MME	0.470427956	0.848684211	0.97368421	0.875
SPARC-ARHCEF10-MANBA	0.558889696	0.842105263	0.84210526	1
SPARC-MANBA-ANXA2	0.551741193	0.842105263	0.84210526	1
BCAR1-DHX9-MANBA	0.538775064	0.837171053	0.86842105	0.96875

**(4) Clinical Implications of Biomarkers for CP treatment and prognosis assessment.**

Although our study primarily aimed to identify early diagnostic biomarkers for CP, **we fully agree with**
**the reviewer’s insightful comments that the identified molecules—though showing modest**
**expression changes—might hold potential clinical value beyond diagnosis.** Given their involvement
in CP-related pathways, these molecules could potentially inform treatment decisions and prognosis
assessment.

The ten diagnostic biomarkers were mainly enriched in extracellular matrix remodeling (e.g., SPARC,
BCAR1), oxidative stress (e.g., LONP1, DHX9) and neuronal signaling (e.g., GNAI3, ANXA2)
pathways, which have been shown to play an important role in neurodevelopmental disorders. For
instance, upregulation of SPARC and BCAR1 implied possible dysregulation of extracellular matrix
dynamics and focal adhesion signaling, which are associated with neuroinflammation and glial cell
activation (*J Neuroinflammation*, 2016, PMID: 27581191), while increased expression of LONP1 might
reflect enhanced mitochondrial stress response, which plays an important role in neurodevelopmental
disorders (*Cell Death Dis*, 2018, PMID: 29899330). **These expression alterations suggested the**
**presence of ongoing pathophysiological processes in neural development and cellular metabolism**
**in CP patients, which could provide a starting point for further investigation into potential**
**therapeutic approaches.** However, it should also be acknowledged that some of these biomarkers may
primarily reflect downstream effects of CP-related pathology. Future research should systematically
evaluate these proteins in relevant cellular and animal models, which might help clarify their functional
significance and contribute to the development of novel strategies for CP.

**In addition, given that serum proteins can dynamically reflect pathophysiological changes,**
**molecular biomarkers might offer value in longitudinal monitoring of disease progression or**
**treatment response.** For instance, alpha-fetoprotein (AFP) is widely used not only as a diagnostic
biomarker for hepatocellular carcinoma but also to monitor post-surgical recurrence (*Nat Rev*
*Gastroenterol Hepatol*, 2022, PMID: 35676420); HMGB1 has been reported as a biomarker not only for
the diagnosis of epilepsy, but also for monitoring treatment response, particularly in relation to
neuroinflammatory activity (*J Clin Invest*, 2017, PMID: 28504645; *Nat Rev Neurol*, 2019, PMID:
31263255). In our study, proteins such as GNAI3 and ANXA2 were characteristically downregulated in
CP patients, potentially reflecting underlying neurodevelopmental impairments. Given their involvement
in neuronal plasticity and cellular communication, it is conceivable that the upregulation of these
molecules might predict a neurorestorative response after treatment (e.g., intrathecal baclofen therapy,
BoNT-A therapy). Although longitudinal samples were not available in the current study to validate this
hypothesis, this limitation will be addressed in the future work by collecting paired pre- and post-
treatment samples. In addition, further mechanistic studies will be necessary to elucidate the functional
roles of these molecules in neurodevelopment and treatment response. These investigations might help
clarify temporal expression patterns of biomarkers and contribute to the development of longitudinal
monitoring strategies for assessing treatment efficacy and prognosis in CP.

In conclusion, the enrichment of these molecules in neurodevelopmentally relevant pathways suggests
that they might also have potential utility in informing future therapeutic research and prognosis

assessment, pending further validation. Moving forward, we aim to refine the biomarker panel through
larger-scale validation, optimize detection methods for clinical implementation, and pursue mechanistic
studies to establish their role in disease modulation. Collectively, these findings provided a promising
basis for the future utility of molecular biomarkers into clinically useful tools for CP management.

**R2Q5. In the third part of the results (line 207), the study states, “we next classified these samples**
**into genetic CP, non-genetic CP, and healthy controls.” What criteria were used for this**
**classification? Please clarify and provide additional details.**

**Response to R2Q5:**

We sincerely apologize for the unclear definitions of the sample classification (genetic CP, non-genetic
CP, and healthy controls) in the original manuscript. **In the revised manuscript**, we have provided a
clearer and more detailed description of these classifications in the **Methods section**. The detailed
responses are as follows.

**(1) As for the definition of genetic CP and non-genetic CP**

All patients underwent comprehensive clinical evaluations conducted by pediatric neurologists, and their
diagnoses conformed to the criteria defined by the Surveillance of Cerebral Palsy in Europe (SCPE) (*Dev*
*Med Child Neurol Suppl*, 2000, PMID: 11132255).

In the original manuscript, the variant results for 321 patients among 346 CP were obtained from our
previously study (*Nat Med*, 2024, PMID: 38693247). **The classification of patients into genetic and**
**non-genetic CP groups was based on the pathogenic and likely pathogenic (P/LP) variants**
**identified in that study**. In response to the reviewer’s comment, we have included a schematic overview
of the variant filtering strategy, as described in the original publication (**Figure RL11**). Briefly, variants
were annotated using ANNOVAR (v2020-06-08). Variants were retained if they were located in genes
listed in the OMIM and HGMD databases and associated with CP phenotypes. These variants were
subsequently categorized as pathogenic (P), likely pathogenic (LP), of uncertain significance (VUS),
benign (B), or likely benign (LB) according to ACMG/AMP guidelines. Because pathogenic and likely
pathogenic (P/LP) variants are considered clinically significant and linked to disease susceptibility
(*Genet Med*, 2015, PMID: 25741868), all such variants identified in the study were confirmed by Sanger
sequencing. **Finally, individuals were classified as genetic CP if they carried at least one P/LP**
**variant in a gene with a known inheritance pattern and a clinical phenotype consistent with the**
**gene’s associated disorder. The classification of non-genetic CP in this study refers to cases in which**
**no pathogenic or likely pathogenic variants were identified.**

**Figure RL11.** Schematic diagram showing the selection process of genetic variants (*Nat Med*, 2024,
 PMID: 38693247).

**(2) As for the definition of Healthy Control.**

Healthy control samples were obtained during routine pediatric medical examinations at the Third
 Affiliated Hospital of Zhengzhou University, a major hospital responsible for standardized health
 screening of infants and children in Henan Province, China. Parents or legal guardians of all children
 included in this study were counseled and provided written informed consent for participation. This
 consent encompassed the use of biological samples for other approved biomedical research. The
 diagnostic criteria were consistent with our previous studies (*Mol Psychiatry*, 2024, PMID: 38762692;
 *Brain Behav Immun*, 2023, PMID: 37011865). Inclusion and exclusion criteria were as follows:

- • **The inclusion criteria:** 1) Typically developing children with no history of neurological,
 genetic, or developmental disorders. 2) No recent perinatal complications or current medications;
- • **The exclusion criteria:** 1) Recent infections, fever, or vaccination within two weeks; 2) First-
 degree family history of neurodevelopmental or inherited genetic disorders;

Thank you again for your professional and constructive comments, which have greatly helped improve
 the quality of this manuscript. The related information was updated in “**Methods**” section (**lines 494-513**
 **on pages 17; lines 524-536 on pages 18**) in the revised manuscript.

**R2Q6. The manuscript mentions that low birth weight or preterm birth is closely associated with**
 **spastic CP. However, given that spastic CP accounts for a large proportion of the cases collected,**
 **could this result in bias in the findings? (line 258, Figure 1B)**

**Response to R2Q6:**

We appreciate the reviewer’s insightful comments regarding the potential bias in our study cohort. To
 answer the reviewer’s questions clearly, we divided the response into two parts: (1) Consistency of cohort

design with epidemiological trends; (2) Impact of spastic CP proportion on study results.

**(1) Consistency of cohort design with epidemiological trends**

CP results from injury to the developing brain caused by diverse etiologies, leading to a wide range of
clinical manifestations and varying degrees of severity. Based on the predominant type of movement
disorder, CP is categorized into spastic, dyskinetic and ataxic CP (*JAMA Pediatr*, 2017, PMID:
28715518). Among these, **spastic CP is the most common, representing approximately 80% of cases**
(*Lancet Neurol*, 2023, PMID: 36657477). Moreover, spastic CP is the main subtype associated with
preterm birth and low birth weight. This might be attributed to the fact that these factors could damage
the corticospinal tracts and periventricular white matter, which are involved in spastic CP
pathophysiology (*Ann Neurol*, 2010 PMID: 20695013; *BMJ*, 2004, PMID: 15591566).

In this study, we collected serum samples from a cohort of 346 patients with CP. The distribution of
clinical subtypes was as follows: spastic (80.8%, n = 280), dyskinetic (4.9%, n = 17), ataxic (2.7%, n =
9), and mixed type (11.6%, n = 39). Notably, the prevalence of low birth weight was significantly higher
among patients with spastic CP (35%) compared to those with non-spastic subtypes (11%). Similarly,
preterm birth was more common in the spastic CP (40%) than in the non-spastic subtypes (12%). **These**
**clinical distributions were in line with previously reported epidemiological trends and suggest that**
**our cohort design was reasonable** (*JAMA Pediatr*, 2017, PMID: 28715518; *Lancet Neurol*, 2023, PMID:
36657477).

**(2) Impact of spastic CP proportion on study results**

**To better answer the reviewer's suggestion, we systematically assessed the impact of subtype**
**distribution on our results in revised version.** Specifically, we constructed a reorganized validation
**cohort by randomly selecting a subset of spastic CP cases (n=66) and matching them with an equal**
**number of non-spastic CP cases (n=66), along with 196 healthy controls (HC) in this version.** To
investigate the subtype-specific associations between perinatal factors (preterm birth and low birth
weight) and CP, **we first evaluated the enrichment of these factors in spastic versus non-spastic CP**
**cases.** Fisher's exact test revealed that both low birth weight and preterm birth remained significantly
enriched in the spastic CP group ($P < 0.05$), suggesting a subtype-specific association that remains
significant after adjusting for the sampling imbalance of spastic CP cases (**Figure RL12A**).

**Next, we performed differential protein expression analysis in the reorganized cohort.** Among the
82 upregulated proteins in original manuscript, **75 (91.5%) remained consistently upregulated**
**(CP/HC > 1.5), with 66 retaining significance after multiple testing correction (FDR < 0.05).**
Similarly, within 83 downregulated proteins in original manuscript, **75 (90.4%) continued to show a**
**consistent downregulation trend (CP/HC < 0.65), of which 69 remained significant (FDR < 0.05)**
**(Figure RL12B).** In addition, the upregulated proteins were significantly enriched in Integrin signaling
(e.g., BCAR1, VWF, FN1), Lysosome (e.g., HEXA, CTSC, AP3D1) and sugar metabolism pathways
(e.g., HK1, MPI) (FDR < 0.05) (**Figure RL12C**). Conversely, proteins decreased were enriched in
calcium signaling (e.g., STIM1, MYLK, GNAS) and thyroid stimulating hormone pathways (e.g.,
APEX1, STAT3) (FDR < 0.05). Notably, the identified proteins and enriched pathways showed substantial

concordance with those reported in the original cohort, indicating high reproducibility and robustness of
 the proteomic signature across varying subtype compositions.

**To assess diagnostic performance across varying subtype distributions, we developed a predictive**
 **model utilizing the same biomarker panel as the original study.** Employing XGBoost algorithm with
 10-fold cross-validation, the model showed robust diagnostic capability in the test set: AUC=0.91,
 accuracy=0.80, precision=0.81, recall=0.80, and F1-score=0.80 (**Figure RL12D**), indicating robust
 diagnostic ability for CP.

The substantial overlap of differentially expressed proteins, and stable performance of the diagnostic
 model in a reorganized validation cohort collectively suggested the robustness and generalizability of
 our findings across CP subtypes, indicating that they were not affected by the proportion of spastic cases
 and might hold translational value for broader clinical application.

Thank you again for your professional and constructive comments, which have greatly helped improve
 the quality of this manuscript. The related information was updated in “**Result**” section (**lines 183-189**
 **on pages 6**) in the revised manuscript.

**Figure RL12.** (A) Stacked bars showing the distribution of the proportions of preterm birth and low birth
 weight in spastic and non-spastic CP. *P* value was computed using Fisher’s exact test. (B) Venn plot of
 upregulated or downregulated proteins identified in both original cohort and reorganized cohort. (C)
 Pathway enrichment analysis of differentially expression proteins (DEPs) illustrating upregulated (red)

and downregulated (blue) signal pathways in CP versus healthy controls. (D) Left: Receiver Operating
Characteristic (ROC) curve depicting the AUC for the XGBoost models. Right: Confusion matrix
showing the performance of the machine learning classifier on the 20% testing set.

**R2Q7. In Figure 5H, what are the precision and recall corresponding to the reported AUC?**

**Response to R2Q7:**

We apologize for any lack of clarity in describing the performance metrics associated with the reported
AUC in Figure 5H. **In the revised manuscript, we added these details, specifying an accuracy of 0.7,**
**a precision of 0.71, a recall of 0.7, and an F1-score of 0.7. These additional metrics offered a more**
**comprehensive evaluation of the model's performance. This information has been incorporated**
**into Figure 5H.**

Thank you again for your professional and constructive comments, which have greatly helped improve
the quality of this manuscript. The related information of data preprocessing was updated in **Figure 5H**
in the revised manuscript.

**Conclusion Section**

**R2Q8. Based on the results in Figure 1H, how can a diagnostic biomarker combination strategy be**
**formulated? Expanding on this point in the discussion section could provide valuable guidance for**
**future research.**

**Response to R2Q8:**

Thank you for your valuable feedback. **we fully agree with the reviewer's insightful comments that**
**formulating diagnostic biomarker combination strategies could provide important guidance for**
**future research.** In clinical practice, single biomarkers offer certain advantages due to their simplicity
in detection and operational convenience (*Sci Transl Med*, 2010, PMID: 20739680). However, due to the
intrinsic limitations of single biomarkers in terms of specificity and stability, the use of multi-biomarker
panels has emerged as a prominent trend in both biomedical research and translational applications (*Nat*
*Biotechnol*, 2006, PMID: 16900146). Nevertheless, **the implementation of such strategies still**
**requires further standardization, primarily in two key areas: the standardized quantification of**
**biomarkers and the establishment of scientifically sound diagnostic thresholds based on their**
**expression levels.**

By leveraging unique peptide mass and fragmentation patterns for protein identification, mass
spectrometry (MS) has become a primary platform for biomarker discovery and detection (*Nat Med*,
2022, PMID: 35654907; *Nat Med*, 2025, PMID: 40164724). **In the original manuscript**, the collected
serum samples were initially analyzed by liquid chromatography-tandem mass spectrometry (LC-
MS/MS). Through standard machine learning procedures, we constructed a multi-marker panel
predictive model that showed strong diagnostic performance, with an AUC of 0.96, accuracy of 0.88,
precision of 0.88, recall of 0.91, and F1-score of 0.88, highlighting its potential clinical utility for CP
diagnosis. To improve the accuracy and reproducibility of biomarker detection, future studies may adopt

stable isotope-labeled standards and establish absolute quantification methods based on mass
spectrometry, thereby promoting their standardized application in clinical settings.

However, given the high cost and operational complexity associated with LC-MS/MS, it is important to
find more cost-effective and technically accessible assays for wider clinical applications. The ELISA
enables absolute quantification of proteins and is widely adopted in routine diagnostics owing to its high
efficiency and cost-effectiveness. Therefore, to validate the cross-platform reproducibility of the
candidate biomarkers, their expression levels were further examined using ELISA in an independent
validation cohort (CP=38, HC=32). The results showed that all 10 biomarkers remained statistically
significant on the ELISA platform ($P < 0.05$) (**Figure RL13A**), indicating potential for clinical utility.

**To better answer the reviewer's question, we further integrated the 10 biomarkers ELISA result**
**into the prediction model, resulting in improved performance (AUC = 0.98; accuracy = 0.93;**
**precision = 0.94; recall = 0.94; F1 score = 0.94) in the revised manuscript (Figure RL13 A-B).**

**To facilitate clinical interpretation and application, defining clear diagnostic thresholds for these**
**biomarkers is critical. We employed a logistic regression model in combination with the Youden**
**index to quantify a diagnostic threshold to enhance the discrimination between CP and control**
**groups in the revised version.** This strategy is consistent with widely adopted approaches in the
literature and offers good generalizability and reproducibility (*Adv Sci*, 2021, PMID: 34026427; *Nat Med*,
2025, PMID: 40164724). Specifically, a logistic regression model was constructed using the ELISA-
derived expression levels of the 10 candidate biomarkers to calculate a risk score for each participant.
The optimal cutoff value was then determined using the Youden index, aiming to achieve the best balance
between sensitivity and specificity.

The composite risk score was calculated using the following formula:

$$\text{Risk Score} = w_0 + w_1 \cdot \text{Protein}_1 + w_2 \cdot \text{Protein}_2 + \dots + w_n \cdot \text{Protein}_n$$

In this formula, w_0 represents the intercept, while w_1, w_2, \dots, w_n are the weighted coefficients
corresponding to each biomarker. $\text{Protein}_1, \text{Protein}_2, \dots, \text{Protein}_n$ represent the log₂-transformed ELISA
expression levels of the 10 biomarkers.

The composite risk score, derived from all 10 biomarkers, exhibited a statistically significant distinction
between CP and healthy controls ($P < 0.05$) (**Figure RL13 C**). **Applying the Youden index, we then**
**established an optimal cutoff value of 0.59 for the 10-biomarker panel.** Samples exceeding this
threshold were classified as CP, while those below were designated as healthy controls. This approach
achieved a sensitivity of 0.95, specificity of 1 and Youden Index of 0.95, demonstrating strong potential
as a diagnostic tool for CP.

**To maintain diagnostic performance while improving cost-efficiency and operational simplicity, we**
**further optimized the model by reducing the number of biomarkers.** Specifically, we systematically
evaluated all possible three-marker combinations (a total of 120 models) and calculated risk scores for
each. Among all 120 three-protein combinations, the panel of SPARC, DHX9, and MANBA achieved
the highest Youden index (0.88), with a sensitivity of 0.95, specificity of 0.94, and an optimal cutoff

value of 0.49 (**Figure RL13D**). This optimized panel model requires only three biomarkers, offering
practical advantages in implementation and broader applicability.

Furthermore, given the limitations of current commercial antibodies—particularly in terms of specificity,
sensitivity, and batch consistency, which are critical in heterogeneous diseases like CP—future efforts
should prioritize the development of high-affinity antibodies and the implementation of large-scale,
multicenter validation studies to support clinical utility. In summary, our identification of a multi-
biomarker panel with stable diagnostic performance, combined with the establishment of a quantifiable
risk scoring system, lays a solid foundation for future validation and clinical implementation.

**In the revision, we have incorporated this discussion into the “Discussion” section (lines 407-427**
**on pages 15). Specifically, as follows:**

“Compared with single biomarkers, multi-marker panels offer notable advantages in improving
diagnostic accuracy, robustness, and generalizability. In this study, we identified ten candidate
biomarkers through MS screening and validated them using ELISA in an independent cohort, ultimately
establishing a multi-marker model with high predictive accuracy. While mass spectrometry remains a
powerful discovery platform, its high cost and technical demands limit its practical use in clinical settings.
To facilitate broader implementation, future efforts may benefit from the development of high-affinity
ELISA antibodies targeting CP-related proteins, which could enhance sensitivity, specificity, and batch-
to-batch consistency for routine diagnostics.

Nevertheless, translating biomarker research into clinical practice remains challenging. A key step toward
clinical utility is the development of standardized and scalable biomarker integration strategies. To this
end, we constructed a logistic regression model using ELISA-derived protein expression levels and
established a quantitative risk scoring system that effectively distinguished CP patients from healthy
controls. Importantly, the application of the Youden index enabled the definition of an optimized
diagnostic threshold, enhancing both interpretability and usability of the model in potential clinical
workflows. In addition, we identified optimized three-protein panel (SPARC, DHX9, and MANBA) that
retained good diagnostic performance while significantly reducing assay complexity. This suggested that
a cost-effective, streamlined implementation of multi-marker strategies may be feasible in routine clinical
practice. The modeling and simplification framework might serve as a useful reference for biomarker
development CP. Future multicenter studies will be essential to validate the clinical utility of this
diagnostic strategy.”

Thank you again for your professional and constructive comments, which have greatly helped improve
the quality of this manuscript.

**Figure RL13.** (A) Boxplot showing the ELISA result of candidate biomarkers. (B) Receiver Operating
 Characteristic (ROC) curve evaluating the performance of biomarkers based on ELISA results in an
 independent cohort, accompanied by a confusion matrix illustrating the performance of the machine
 learning classifier. (C) Boxplot displaying the risk score distribution based on the 10 biomarkers derived
 from ELISA results. (D) Boxplot showing the risk score based on SPARC-DHX9-MANBA panel.

**R2Q9.** In the final conclusion, the statement “The predicted model based on serum proteins could
 distinguish between different GMFCS levels” seems questionable. Is this claim an overstatement
 of the study’s findings? In the fifth part of the results, it was only reported that high GMFCS levels
 are closely associated with specific IGHV-immunoglobulins, but no specific differential proteins
 were identified across the various GMFCS levels.

**Response to R2Q9:**

We sincerely thank the reviewer’s insightful comment regarding the overstatement of GMFCS-level
 discrimination in our original conclusion. We fully agree that the claim “distinguish between different
 GMFCS levels” was imprecise, as our study focused on binary classification (high vs. low severity)
 rather than multi-class discrimination across all five GMFCS levels.

In the original manuscript, GMFCS levels were binarized (Levels I–III = low severity; Levels IV–V =
 high severity) in line with a previous study (*Brain*, 2022, PMID: 34077496). The 10-protein panel (e.g.,

EXOG, IGHV6-1, RPL23) was selected through XGBoost-based feature importance and achieved
moderate performance (AUC: 0.76) with with 70% accuracy, 71% precision, 70% recall, and an F1-score
of 70%. These results suggested moderate discriminative ability for distinguishing between high and low
severity CP patients.

**We apologize for this oversight and have revised the conclusion accordingly. The revised statement:**
**“(4) Serum proteomics might have potential in assessing CP severity, particularly as IGHV-related**
**immunoglobulins were positively correlated with GMFCS level.”** Thank you again for your
professional and constructive comments, which have greatly helped improve the quality of this
manuscript. The related information was updated in **“Discussion” section (lines 471-473 on pages 16)**
**in the revised manuscript.**

**Methods Section**

**R2Q10. Regarding the collection of blood samples, how much blood was drawn from each**
**participant? What were the components of the collection tubes? These details are crucial for**
**ensuring the quality of the proteomic analysis.**

**Response to R2Q10**

We sincerely thank the reviewer for this valuable comment. Standardized serum sample collection and
processing are essential for ensuring high-quality proteomic data. In this study, **all samples were**
**obtained by certified clinicians at affiliated hospitals of Zhengzhou University with informed**
**consent, and were collected and processed in strict accordance with the Guidelines for Venous**
**Blood Specimen Collection (WS/T 661-2020).** Specifically:

- • **Collection tubes:** The blood collection was performed using standard yellow-top vacutainer
**tubes with GEL and clot activator**, which are specifically designed for serum collection
(Shandong Acealife Medical Devices Co., Ltd, China).
- • **Sample Volume: approximately 1–2 mL of peripheral venous blood was obtained per**
**participant.**
- • **Sample Processing and Storage:** The whole blood samples were allowed to clot at room temperature
and subsequently centrifuged at 1,000 × g for 10 minutes at 4 °C to separate the serum. All samples
were processed within 2 hours of collection, and the serum was aliquoted into 2.0 mL sterile
cryovials (Corning®, Cat. No. 430659) and stored at –80 °C until further analysis.

This standardized protocol is widely adopted in large-scale proteomic studies and effectively reduce the
impact of long-term storage on sample stability and data reliability (*EMBO Mol Med*, 2022, PMID:
34978375; *Cell*, 2020, PMID: 32492406).

Thank you again for your professional and constructive comments, which have greatly helped improve
the quality of this manuscript. The related information was updated in **“Methods” section (lines 516-521**
**on pages 17-18)** in the revised manuscript.

**R2Q11. It is well known that obtaining samples from healthy controls is more challenging**
**compared to CP patients. The methods section mentions that the control group samples were**
**collected from healthy children in kindergartens. How were these children recruited? Were there**
**specific inclusion/exclusion criteria for the CP and control groups? The details provided in the**
**manuscript are limited; please supplement this information.**

**Response to R2Q11**

We thank the reviewer for this important comment and apologize for the lack of detail in the original
manuscript. **In response, we have updated the Methods section to provide a clearer description of**
**the recruitment procedures and inclusion/exclusion criteria for both the CP and control groups.**

**(1) About the recruitment of the healthy control group**

Healthy control samples were obtained during routine pediatric medical examinations at the Third
Affiliated Hospital of Zhengzhou University, a major hospital responsible for standardized health
screening of infants and children in Henan Province, China. As part of standard care, **Parents or legal**
**guardians of all children included in this study were counseled and provided written informed**
**consent for participation.** This consent encompassed the use of biological samples for other approved
biomedical research.

Serum samples from the control group were collected, processed, and stored following the same
standardized protocol as the CP group (see **R2Q10**) to ensure analytical consistency.

**(2) About the inclusion and exclusion criteria for the CP and control groups.**

The diagnostic criteria for CP were consistent with our previous study (*Nat Med*, 2024, PMID: 38693247).

**All children diagnosed with CP met Surveillance of Cerebral Palsy in Europe (SCPE) criteria (*Dev***
***Med Child Neurol Suppl*, 2000, PMID: 11132255).** Inclusion and exclusion criteria were as follows:

- • **The inclusion criteria:** 1) All patients were under 18 years of age. 2) Diagnosed with CP following
comprehensive clinical evaluations conducted by pediatric neurologists, including pregnancy and
birth history, family history, and detailed physical and neurological examinations; 3) No known
chromosomal abnormalities or syndromic diagnoses.
- • **The exclusion criteria:** 1) Unstable vital signs or serious systemic illness; 2) CP secondary to
acquired causes (e.g., encephalitis, uncontrolled epilepsy); 3) CP subtypes predominantly
characterized by hypotonia or rigidity.

**For healthy controls,** the diagnostic criteria were consistent with our previous studies (*Mol Psychiatry*,
2024, PMID: 38762692; *Brain Behav Immun*, 2023, PMID: 37011865). All participants were pre-
screened using structured questionnaires and underwent clinical evaluations by qualified pediatricians to
ensure eligibility and reduce potential confounding. Inclusion and exclusion criteria were as follows:

- • **The inclusion criteria:** 1) **Typically developing children with no history of neurological,**
**genetic, or developmental disorders.** 2) No recent perinatal complications or current medications;
- • **The exclusion criteria:** 1) Recent infections, fever, or vaccination within two weeks; 2) First-
degree family history of neurodevelopmental or inherited genetic disorders;

Thank you again for your professional and constructive comments, which have greatly helped improve
the quality of this manuscript. The related information was updated in “**Methods**” section (**lines 491-513**
**on pages 18**) in the revised manuscript.

**R2Q12. In attachment 9836907 sheet1, two samples lack GMFCS classification and other related**
**information. Were these samples excluded from the analysis? Additionally, no data on CP patients**
**is present in sheet7. Furthermore, in attachment 9836908 sheet1, the number of P/LP cases does**
**not match the 76 cases mentioned in the manuscript. Please clarify these inconsistencies.**

**Response to R2Q12:**

Thank you for your valuable feedback. We apologize for any confusion caused by the unclear description
in the original manuscript. The specific clarifications were as follows:

**(1) About two samples lack GMFCS classification and other related information.**

The two CP samples with missing clinical information were included only in Figure 1 for analyzing
protein-level differences between disease and healthy groups. These samples were excluded from all
subsequent analyses.

**(2) About CP patients present in sheet7.**

We sincerely apologize for the missing patient information. The sheet7 of attachment 9836907 contains
the ELISA results from the validation cohort. **In the revised Supplementary Table 1, we have updated**
**the corresponding basic clinical information for these samples.**

**(3) About the number of P/LP cases and P/LP variants.**

We sincerely appreciate the reviewer’s careful assessment and the opportunity to clarify this
inconsistency. After re-evaluating our data, **65 patients in our cohort were identified as carriers of**
**pathogenic or likely pathogenic (P/LP) variants.** However, **a subset of these patients harbored more**
**than one P/LP variant.** As a result, **the number “76” reported in the manuscript refers to the total**
**number of P/LP variants, not the number of carriers.** We apologize for the confusion and have
clarified this distinction in the revised version.

In response to the reviewer’s valuable feedback. The related information was updated in “**Results**”
section (**lines 224-225 on pages 8**) in the revised manuscript.

**R2Q13. Regarding the manuscript's writing, there is room for improvement. For example, in line**
**114, the first occurrence of an English abbreviation should include its full form. There is also an**
**abbreviation error on line 279. Some expressions are ambiguous, such as “Nevertheless, its**
**application in CP remains underexplored, warranting further research to validate its role as a**
**diagnostic tool.” The serum proteomic is a method for discovering diagnostic biomarkers, not a**
**diagnostic tool for the disease itself—diagnostic biomarkers are. The manuscript requires further**
**revisions to enhance its clarity and readability.**

**Response to R2Q13:**

We sincerely thank the reviewer for the careful reading and constructive suggestions regarding the clarity
and readability of the manuscript. In the revised version, we have made substantial improvements in
writing style and English language proficiency across the entire document. We have thoroughly checked
the manuscript to correct grammatical and spelling errors. The revised manuscript was sent to
professional science and native speaker writers for their suggestions and editing. **All changes are**
**highlighted in red in the revised manuscript.**

1) About the abbreviations: We have carefully reviewed all abbreviations in the revised manuscript. The
full term of each abbreviation (e.g., “**weighted gene co-expression network analysis (WGCNA)**” in
line 113) has been provided upon first occurrence, and the abbreviation error identified has been corrected
(e.g., “**gestational age (GA)**” in line 319).

2) About the ambiguous expression: We agree with the reviewer that the original sentence, “its
application in CP remains underexplored, warranting further research to validate its role as a diagnostic
tool,” may lead to a misunderstanding. As suggested, serum proteomics is a discovery platform rather
than a diagnostic tool itself. We have revised the sentence accordingly to clarify its role in biomarker
discovery:

“**Serum biomarkers are capable of detecting disease-specific alterations and have shown diagnostic**
**utility across various neurological disorders. However, their application in CP remains limited,**
**highlighting the need for further studies to identify and validate serum biomarkers for early**
**diagnosis.” (in line 85-88)**

**R2Q14. The references section lacks consistent formatting and requires standardization.**

**Response to R2Q14:**

We thank the reviewer for the helpful comment and sincerely apologize for the inconsistency in reference
formatting. **In the revised manuscript, all references have been reformatted in accordance with the**
***Nature communication* journal guidelines to ensure accuracy and consistency.**
